# Accelerated Methods with Complexity Separation Under Data Similarity for Federated Learning Problems

## Abstract

Heterogeneity within data distribution poses a challenge in many modern federated learning tasks. We formalize it as an optimization problem involving a computationally heavy composite under data similarity. By employing different sets of assumptions, we present several approaches to develop communication-efficient methods. An optimal algorithm is proposed for the convex case. The constructed theory is validated through a series of experiments across various problems.

## 1 Introduction

Currently, the field of optimization theory is well-developed. It includes a wide range of algorithms and techniques designed to efficiently solve various tasks. In today's landscape, engineers often have to handle large-scale data. It can be spread across multiple nodes/clients/devices/machines to share the load by working in parallel (Verbraeken et al., 2020). The problem can be formally written as

$$\min_{x \in \mathbb{R}^d} \left[ h(x) = \frac{1}{|M_h|} \sum_{m \in M_h} h_m(x) \right], \quad \text{with } h_m(x) = \frac{1}{n_m} \sum_{j=1}^{n_m} \ell(x, z_j^m), \tag{1}$$

where $n_m$ is the size of the $m$-th local dataset, $x$ is the vector of model parameters, $z_j^m$ is the $j$-th data point of the $m$-th dataset, and $\ell$ is the loss function. The most computationally powerful device ($h_1$, without loss of generality) is treated as a server, while the others communicate through it.

The primary challenge that must be addressed in this paradigm is a communication bottleneck (Jordan et al., 2019). Deep models are often extremely large, and excessive information exchange can negate the acceleration provided by computational parallelism (Kairouz et al., 2021). A potential solution to reduce the frequency of communication is to exploit the similarity of local data. There are several ways to measure this phenomenon. The most mathematically solid one typically employs the Hessians.

**Definition 1.** *(Hessian similarity). We say that $h_i$, $h_j$ are $\delta$-related, if there exists a constant $\delta > 0$ such that*

$$\|\nabla^2 h_i(x) - \nabla^2 h_j(x)\| \leq \delta, \quad \forall x \in \mathbb{R}^d.$$

Many papers assume the relatedness of every $h_m$ and $h$ (Lin et al., 2024; Jiang et al., 2024), but we rely only on $h_1$ and $h$, as in (Shamir et al., 2014; Hendrikx et al., 2020; Kovalev et al., 2022). If we consider $h$ to be $L$-smooth, the losses exhibit greater statistical similarity with the growth of the local dataset size $n$. Measure concentration theory implies $\delta \sim {}^L/n$ and $\delta \sim {}^L/\sqrt{n}$ in the quadratic and general cases, respectively (Hendrikx et al., 2020). In distributed learning, where samples are shared between machines manually, it is easy to produce a homogeneous distribution. As a result, schemes using $\delta$-relatedness strongly outperform their competitors.

However, new settings entail new challenges. Federated learning (Zhang et al., 2021) requires working with private data that is collected locally by clients and may therefore be heterogeneous. As a result, similarity-based methods lose quality (see Table 3 in (Karimireddy et al., 2020)). We argue that this issue can be addressed to some extent. In practice, some distribution modes are common and shared uniformly between the server and the clients, while others are unique and primarily contained on the devices. In that case, one part of the data is better approximated by the server than the other. This suggests the idea that the objective can be represented as a sum of two functions,

corresponding to frequent and rare data. The new problem can be formulated as

$$\min_{x \in \mathbb{R}^d} \left[ h(x) = \frac{1}{|M_f|} \sum_{m \in M_f} f_m(x) + \frac{1}{|M_g|} \sum_{m \in M_g} g_m(x) \right], \tag{2}$$

where $M_f$, $M_g$ denotes the set of devices sharing $f$, $g$, respectively, and $|\cdot|$ is the number of nodes in the corresponding set. $f$ and $g$ are the empirical losses corresponding to common and rare modes, respectively. The problem 2 exhibits a composite structure. It consists of components that are distinct from one another, including in terms of similarity. For the most typical samples, the server ($h_1 = f_1 + g_1$) may possess more extensive information, while for unique instances, there may be no ability to reproduce them accurately. In this context, we propose to examine two characteristics of relatedness simultaneously:

$$\|\nabla^2 f_1(x) - \nabla^2 f(x)\| \le \delta_f, \quad \forall x \in \mathbb{R}^d;$$
$$\|\nabla^2 g_1(x) - \nabla^2 g(x)\| \le \delta_g, \quad \forall x \in \mathbb{R}^d.$$

Without loss of generality, we assume $\delta_f < \delta_g$. Due to the additional structure of the objective, it is possible to call $f$ (and hence the devices from $M_f$) less frequently. Thus, communication bottleneck can be addressed more effectively than SOTA approaches suggest. Indeed, the complexity of existing schemes depends on $\max\{\delta_f, \delta_g\} = \delta_g$ (Hendrikx et al., 2020; Kovalev et al., 2022; Beznosikov et al., 2024; Lin et al., 2024; Bylinkin and Beznosikov, 2024). This indicates that no account is taken of the fact that certain data modes are better distributed between the server and the clients than others. This presents a unique challenge that requires the development of new schemes. Our paper answers the question:

*How to bridge the gap between separating the complexities in the problem (2) and the Hessian similarity?*

## 2 RELATED WORKS

### 2.1 COMPOSITE OPTIMIZATION

Classic works on numerical methods considered minimizing $h$ without assuming any additional structure (Polyak, 1987). Influenced by the development of machine learning, a composite setting with $h(x) = f(x) + g(x)$ as an objective has emerged. This focused on proximal friendly $g$ (regularizer) (Parikh et al., 2014). This means that any optimization problem over $g$ is easy to solve since its value and gradient are "free" to compute. However, many practical tasks do not satisfy this property. Consequently, the community has shifted towards analyzing more specific scenarios, leading to the emergence of the heavy composite setting. Juditsky et al. (2011) and Lan (2012) studied convex smooth+non-smooth problems but were unable to separate the complexities. The result was $\mathcal{O}\left(\sqrt{L_f/\varepsilon} + L_g^2/\varepsilon^2\right)$. It cannot be improved if only the first-order information of $f + g$ is accessible. However, it is reasonable to expect that the number of $\nabla f$ evaluations can be bounded by $\mathcal{O}\left(\sqrt{L_f/\varepsilon}\right)$ if the non-smooth term $g$ is absent. This suggests that the estimate can be enhanced if there is separate access to the first-order information of $f$ and $g$. A step in this direction was taken with the invention of gradient sliding in (Lan, 2016). For convex $f$ and $g$, the author managed to obtain $\mathcal{O}\left(\sqrt{L_f/\varepsilon}\right)$ and $\mathcal{O}\left(\sqrt{L_f/\varepsilon} + L_g^2/\varepsilon^2\right)$ of $\nabla f$ and $g' \in \partial g$ evaluations, respectively. Later, the exact separation was achieved for convex smooth+smooth problems in Lan and Ouyang (2016). The proposed method achieved $\mathcal{O}\left(\sqrt{L_f/\varepsilon}\right)$ and $\mathcal{O}\left(\sqrt{L_g/\varepsilon}\right)$. For strongly convex $f$, $g$, the result was $\mathcal{O}\left(\sqrt{L_f/\mu} \log 1/\varepsilon\right)$ and $\mathcal{O}\left(\sqrt{L_g/\mu} \log 1/\varepsilon\right)$.

At present, various exotic sliding-based schemes exist: for VIs (Lan and Ouyang, 2021; Emelyanov et al., 2024), saddle points (Tominin et al., 2021; Kuruzov et al., 2022; Borodich et al., 2023), zero-order optimization problems (Beznosikov et al., 2020; Stepanov et al., 2021; Ivanova et al., 2022), and high-order minimization (Kamzolov et al., 2020; Gasnikov et al., 2021; Grapiglia and Nesterov, 2023).

Based on the above literature review, it can be concluded that the concept of complexity separation is well established. Moreover, the sliding approach is utilized to design communication-efficient algorithms based on similarity. The following subsection is dedicated to this topic.

## 2.2 SIMILARITY

The essence of most techniques for handling the Hessian similarity lies in artificially dividing the objective into two components:

$$h(x) = (h - h_1)(x) + h_1(x),$$

where $h - h_1$ is $\delta$-smooth. Unfortunately, previous gradient sliding methods cannot be easily adapted to this setting, as classic works assume the convexity of both components. This presents a challenge in developing a theory for the *convex+non-convex=convex* case.

The first approach addressing similarity was the Newton-type method, DANE, designed for quadratic strongly convex functions (Shamir et al., 2014). For this class of problems, Arjevani and Shamir (2015) established a lower bound on the required number of communication rounds. However, DANE failed to achieve it, prompting the question of how to bridge the gap. Numerous papers explored this issue but either fell short of meeting the exact bound or required specific cases and unnatural assumptions (Zhang and Lin, 2015; Lu et al., 2018; Yuan and Li, 2020; Beznosikov et al., 2021; Tian et al., 2022). Recently, Accelerated ExtraGradient, achieving optimal round complexity, was introduced by Kovalev et al. (2022).

The current trend in this area is to combine similarity with other approaches. This is often non-trivial and demands the development of new techniques. In constructing a scheme with local steps, the theory of the $\delta$-relatedness was first utilized by Karimireddy et al. (2020). The proposed method experienced acceleration due to the similarity of local data, but only for quadratic losses. This result has recently been revisited and significantly improved in (Luo et al., 2025).

Khaled and Jin (2022) attempted to utilize client sampling in the similarity scenario. However, the analysis of the proposed scheme requires strong convexity of each local function, which sufficiently narrows the class of problems. Moreover, the authors used CATALYST (Lin et al., 2015) to accelerate the method, which resulted in extra logarithm multiplier in the complexity and experimental instability. This issue was addressed in (Lin et al., 2024). AccSVRS achieved an optimal number of client-server communications.

Combining compression and similarity is also widespread in research papers. One of the first results in this area was obtained by Beznosikov and Gasnikov (2022). The authors proposed schemes utilizing both unbiased and biased compression. However, the complexity includes a term that depends on the Lipschitz constant of the objective's gradient. This issue was addressed in (Beznosikov et al., 2024), but only for the permutation compression operator. Recently, similarity and compression (both unbiased and biased) have been combined in an accelerated method designed by Bylinkin and Beznosikov (2024).

**Similarity + composite structure of the objective** represents an interesting challenge that has not been addressed.

## 3 OUR CONTRIBUTION

We analyze the problem 2 under the Hessian similarity condition. This paper presents several efficient methods for various sets of assumptions.

• Firstly, we consider the setting of strongly convex $h$ and possibly non-convex $f$, $g$. Starting with a naive stochastic approach, we construct a method with exact separation of complexities: $\mathcal{O}\left(\sqrt{\delta_f/\mu} \log 1/\varepsilon + \sigma^2/\mu\varepsilon\right)$ and $\mathcal{O}\left(\sqrt{\delta_g/\mu} \log 1/\varepsilon + \sigma^2/\mu\varepsilon\right)$ communication rounds for the nodes from $M_f$ and $M_g$, respectively. It is not optimal because of sublinear terms in the estimates. To address this issue, we develop the variance reduction theory for the problem 2. Overcoming several challenges, we present **V**ariance **R**eduction for **C**omposite under **S**imilarity (VRCS) that achieves $\mathcal{O}\left(\delta_f/\mu \log 1/\varepsilon\right)$ and $\mathcal{O}\left((\delta_g/\delta_f)\delta_g/\mu \log 1/\varepsilon\right)$. Its accelerated version AccVRCS enjoys $\mathcal{O}\left(\sqrt{\delta_f/\mu} \log 1/\varepsilon\right)$ and $\mathcal{O}\left((\delta_g/\delta_f)^{3/2}\sqrt{\delta_g/\mu} \log 1/\varepsilon\right)$. In summary, we manage to achieve complexity separation with the optimal estimate for $M_f$ and the extra factor $(\delta_g/\delta_f)^{3/2}$ for $M_g$. To make both complexities optimal, we have to impose requirements on $g$, see the following paragraph.

• Under the additional assumption of $g$ convexity, we propose an approach based on Accelerated Extragradient. Our method enjoys separated communication complexities. It achieves optimal $\mathcal{O}\left(\sqrt{\delta_f/\mu} \log 1/\varepsilon\right)$, $\tilde{\mathcal{O}}\left(\sqrt{\delta_g/\mu} \log 1/\varepsilon\right)$ for $M_f$, $M_g$, respectively.

• We validate our theory through experiments across a diverse set of tasks. Specifically, we evaluate the performance of a *Multilayer Perceptron (MLP)* on the *MNIST* dataset and *ResNet-18* on *CIFAR-10*.

## 4 SETTING

### 4.1 NOTATION

We assume that the devices and their communication channels are equivalent if they belong to the same set of nodes ($M_f$ or $M_g$). In a synchronous setup, analyzing the complexity in terms of communication rounds for $M_f$ and $M_g$ separately is sufficient. The number of times the server initiates communication is considered in this case. This approach does not take into account the number of involved machines and is well-suited for networks with synchronized nodes of two types. When discussing our results, we also utilize the number of communications. This metric counts each client-server vector exchange as a separate unit of complexity and is more appropriate for the asynchronous case.

### 4.2 ASSUMPTIONS

The first part of our work relies solely on standard assumptions, leaving $f$, $g$ arbitrary:

**Assumption 1.** $h \colon \mathbb{R}^d \to \mathbb{R}$ *is $\mu$-strongly convex on $\mathbb{R}^d$:*

$$h(x) \geq h(y) + \langle \nabla h(y), x - y \rangle + \frac{\mu}{2} \|x - y\|^2, \quad \forall x, y \in \mathbb{R}^d. \tag{3}$$

**Assumption 2.** *$f_1$ is $\delta_f$-related to $f$, and $g_1$ is $\delta_g$-related to $g$ (Definition 1). We assume $\mu < \delta_f < \delta_g$.*

Strong convexity of the objective (with arbitrary $f$ and $g$) is the common assumption. No paper on data similarity is void of it (Hendrikx et al., 2020; Kovalev et al., 2022; Beznosikov et al., 2024; Lin et al., 2024; Bylinkin and Beznosikov, 2024). The $\delta$-relatedness does not diminish the generality of our analysis, as in the case of absolutely heterogeneous data, it suffices to substitute $\delta_f = L_f$ and $\delta_g = L_g$ in the results.

Further, we strengthen the setting by assuming $g$ to be convex (only in Section 7):

**Assumption 3.** *$g \colon \mathbb{R}^d \to \mathbb{R}$ is convex ($\mu = 0$) on $\mathbb{R}^d$.*

This allows us to obtain optimal estimates for communication rounds over $M_f$ and $M_g$ simultaneously.

## 5 COMPLEXITY SEPARATION VIA SGD

To construct a theory suitable for applications, we should avoid introducing excessive requirements. Firstly, we analyze the problem (2) without imposing additional conditions on $f$, $g$.

We begin with a naive SGD-like approach (Robbins and Monro, 1951). In Line 5 of Algorithm 1, we propose selecting which part of the nodes ($M_f$ or $M_g$) to communicate with at each iteration. Moreover, we aim to perform sampling based on the similarity constants rather than uniformly. To maintain an unbiased estimator, we normalize it by the probability of choice (Line 5). Additionally, we apply the same scheme in Line 7. Thus, each round of communication involves clients from either $M_f$ or $M_g$.

As previously stated, the stochastic oracles $\xi_k$ and $\zeta_k$ are unbiased. As usual in `SGD`-like algorithms, we impose a variance boundedness assumption to prove the convergence of `SC-AccExtragradient` (Algorithm 1).

**Assumption 4.** *The stochastic oracles $\xi_k$, $\zeta_k$ have bounded variances:*

$$\mathbb{E}_{\xi_k} \left[ \|\xi_k - \nabla(h - h_1)(\underline{x}_k)\|^2 \right] \leq \sigma^2, \quad \mathbb{E}_{\zeta_k} \left[ \|\zeta_k - \nabla h(\overline{x}_{k+1})\|^2 \right] \leq \sigma^2.$$

This approach enables the separation of complexities without introducing assumptions regarding the composite.

**Theorem 1.** *Consider Algorithm 1 for the problem 2 under Assumptions 1-4. Let the subproblem in Line 6 be solved approximately:*

$$\mathbb{E} \left[ \|\nabla A_\theta^k(\overline{x}_{k+1})\|^2 \right] \leq \mathbb{E} \left[ \frac{1}{11\theta^2} \|\underline{x}_k - \arg\min_{x \in \mathbb{R}^d} A_\theta^k(x)\|^2 \right]. \tag{4}$$

---

**Algorithm 1** SC-AccExtragradient

---

1: **Input:** $x_0 = \overline{x}_0 \in \mathbb{R}^d$
2: **Parameters:** $\tau \in (0, 1), \ \eta, \theta, \alpha, p, K > 0$
3: **for** $k = 0, 1, \ldots, K - 1$ **do**
4:      $\underline{x}_k = \tau x_k + (1 - \tau)\overline{x}_k$
5:      $\xi_k = \begin{cases} \frac{1}{p}\nabla(f - f_1)(\underline{x}_k), & \text{with probability } p \\ \frac{1}{1-p}\nabla(g - g_1)(\underline{x}_k), & \text{with probability } 1 - p \end{cases}$

6:      $\overline{x}_{k+1} \approx \arg\min_{x \in \mathbb{R}^d} \left[ A_\theta^k(x) \right]$, where
$$A_\theta^k(x) = \langle \xi_k, x \rangle + \frac{1}{2\theta}\|x - \underline{x}_k\|^2 + h_1(x)$$

7:      $\zeta_k = \begin{cases} \frac{1}{p}\nabla f(\overline{x}_{k+1}), & \text{with probability } p \\ \frac{1}{1-p}\nabla g(\overline{x}_{k+1}), & \text{with probability } 1 - p \end{cases}$
8:      $x_{k+1} = x_k + \eta\alpha(\overline{x}_{k+1} - x_k) - \eta\zeta_k$
9: **end for**
10: **Output:** $x_K$

---

*Then the complexities in terms of communication rounds are*

$$\mathcal{O}\left( \sqrt{\frac{\delta_f}{\mu}} \log \frac{1}{\varepsilon} + \frac{\sigma^2}{\mu\varepsilon} \right), \quad \mathcal{O}\left( \sqrt{\frac{\delta_g}{\mu}} \log \frac{1}{\varepsilon} + \frac{\sigma^2}{\mu\varepsilon} \right)$$

*for the nodes from $M_f$, $M_g$ respectively.*

See the proof in Appendix B.

### 5.1 DISCUSSION

The naive stochastic approach yields a complexity separation without imposing requirements on the problem components. However, the estimates are not optimal and rely on an unnatural Assumption 4. The next step is to remove it by incorporating variance reduction into the proposed algorithm. This constitutes the primary theoretical challenge of our paper.

## 6 COMPLEXITY SEPARATION VIA VARIANCE REDUCTION

To achieve convergence without the sublinear terms, we require Assumptions 1-2 only. We refer to (Lin et al., 2024) that successfully implemented variance reduction for the problem 1. Their SVRG-like gradient estimator does not account for both similarity constants and does not allow for splitting the complexities. Following the logic, we propose to replace $h_1(x_k)$ by $h_1^{i_k}(x_k) - h_1^{i_k}(w_0) + h_1(w_0)$ (see Line 8 of Algorithm 2). We also suggest to use sampling from Bernoulli distribution, same as in Algorithm 1.

By overcoming the technical challenges associated with selecting the appropriate geometry to separate the complexities, we derive the result.

**Theorem 2.** *Consider Algorithm 2 for the problem 2 under Assumptions 1-2. Let the subproblem in Line 9 be solved approximately:*

$$\mathbb{E}\left[ \|\nabla A_\theta^t(x_{t+1})\|^2 \right] \leq \mathbb{E}\left[ \frac{\mu}{17\theta}\|x_t - \arg\min_{x \in \mathbb{R}^d} A_\theta^t(x)\|^2 \right]. \tag{5}$$

*Then the complexities in terms of communication rounds are*

$$\mathcal{O}\left( \frac{\delta_f}{\mu} \log \frac{1}{\varepsilon} \right), \quad \mathcal{O}\left( \left(\frac{\delta_g}{\delta_f}\right) \frac{\delta_g}{\mu} \log \frac{1}{\varepsilon} \right)$$

*for the nodes from $M_f$, $M_g$, respectively.*

See the proof in Appendix D. Due to the difference between $\delta_f$ and $\delta_g$, the number of communication rounds over $M_f$ is reduced. This effect is not "free", since the complexity over $M_g$ is increased by the same number of times.

---

**Algorithm 2** $\mathrm{VRCS}^{\mathrm{1ep}}(p, q, \theta, x_0)$

---

1: **Input:** $x_0 \in \mathbb{R}^d$
2: **Parameters:** $p, q \in (0, 1)$, $\theta > 0$
3: $T \sim \mathrm{Geom}(q)$
4: **for** $t = 0, \ldots, T - 1$ **do**
5: $\quad i_t \sim \mathrm{Be}(p)$
6: $\quad \xi_t = \begin{cases} \frac{1}{p} \nabla(f - f_1)(x_t) \text{ if } i_t = 1, \\ \frac{1}{1-p} \nabla(g - g_1)(x_t) \text{ if } i_t = 0 \end{cases}$
7: $\quad \zeta_t = \begin{cases} \frac{1}{p} \nabla(f - f_1)(x_0) \text{ if } i_t = 1, \\ \frac{1}{1-p} \nabla(g - g_1)(x_0) \text{ if } i_t = 0 \end{cases}$
8: $\quad e_t = \xi_t - \zeta_t + \nabla h(x_0) - \nabla h_1(x_0)$
9: $\quad x_{t+1} \approx \arg\min_{x \in \mathbb{R}^d} [A_\theta^t(x)]$, where

$$A_\theta^t(x) = \langle e_t, x \rangle + \frac{1}{2\theta} \|x - x_t\|^2 + h_1(x)$$

10: **end for**
11: **Output:** $x_T$

---

Next, we utilize an interpolation framework inspired by KatyushaX (Allen-Zhu, 2018) to develop an accelerated version of Algorithm 2. Note that the subproblem appearing in Line 8 of Algorithm

---

**Algorithm 3** AccVRCS

---

1: **Input:** $z_0 = y_0 \in \mathbb{R}^d$
2: **Parameters:** $p, q, \tau \in (0, 1)$, $\theta, \alpha > 0$
3: **for** $k = 0, 1, 2, \ldots, K - 1$ **do**
4: $\quad x_{k+1} = \tau z_k + (1 - \tau) y_k$
5: $\quad y_{k+1} = \mathrm{VRCS}^{\mathrm{1ep}}(p, q, \theta, x_{k+1})$
6: $\quad t_k = \nabla(h - h_1)(x_{k+1}) - \nabla(h - h_1)(y_{k+1})$
7: $\quad G_{k+1} = q\left(t_k + \frac{x_{k+1} - y_{k+1}}{\theta}\right)$
8: $\quad z_{k+1} = \arg\min_{z \in \mathbb{R}^d} q(z)$, where

$$q(z) = \frac{1}{2\alpha} \|z - z_k\|^2 + \langle G_{k+1}, z \rangle + \frac{\mu}{4} \|z - y_{k+1}\|^2$$

9: **end for**
10: **Output:** $y_K$

---

3 can be solved analytically. Therefore, it does not require any additional heavy computations. We provide the convergence result for AccVRCS (Algorithm 3).

**Theorem 3.** *Consider Algorithm 3 for the problem 2 under Assumptions 1-2. Then the complexities in terms of communication rounds are*

$$\mathcal{O}\left(\sqrt{\frac{\delta_f}{\mu}} \log \frac{1}{\varepsilon}\right), \quad \mathcal{O}\left(\left(\frac{\delta_g}{\delta_f}\right)^{3/2} \sqrt{\frac{\delta_g}{\mu}} \log \frac{1}{\varepsilon}\right)$$

*for the nodes from $M_f$, $M_g$, respectively.*

See the proof in Appendix E. As can be seen from Theorem 3, the acceleration has to be paid for by increasing the factor in one of the complexities to $(\delta_g/\delta_f)^{3/2}$.

## 6.1 DISCUSSION

Using only general assumptions, we construct the method that achieves the lower bound on the number of communication rounds across $M_f$. Without imposing additional conditions on the composites, achieving complexities independent of the $\delta_g/\delta_f$ factor is not possible. Nevertheless, the proposed approach is notable for the presence of parameters $p$, $q$, which allow adjustment of the

proportion between communication over $M_f$ and $M_g$. The next chapter addresses the reachability of exact complexity separation.

## 7 COMPLEXITY SEPARATION VIA ACCELERATED EXTRAGRADIENT

In this section, we move on to the more straightforward case, which requires $g$ to be "good" enough. This allows for the adaptation of an already existing technique to yield a satisfying result. For the problem 1, the optimal communication complexity is achieved by `Accelerated Extragradient` (Kovalev et al., 2022).

---

**Algorithm 4** `C-AccExtragradient`

---

1: **Input:** $x_0 = \overline{x}_0 \in \mathbb{R}^d$
2: **Parameters:** $\tau \in (0,1)$, $\eta, \theta, \alpha, K > 0$
3: **for** $k = 0, 1, \dots, K-1$ **do**
4: $\quad \underline{x}_k = \tau_f x_k + (1 - \tau_f)\overline{x}_k$
5: $\quad \overline{x}_{k+1} \approx \arg\min_{x \in \mathbb{R}^d} \left[ A_{\theta_f}^k(x) \right]$, where

$$A_\theta^k(x) = \langle \nabla(f - f_1)(\underline{x}_k), x \rangle + \frac{1}{2\theta_f} \|x - \underline{x}_k\|^2 + f_1(x) + g(x)$$

6: $\quad x_{k+1} = x_k + \eta_f \alpha_f (\overline{x}_{k+1} - x_k) - \eta_f \nabla h(\overline{x}_{k+1})$
7: **end for**
8: **Output:** $x_K$

---

In the first phase, it is proposed to use only the $\delta_f$-relatedness of $f$ and $f_1$, and to place $g$ in the subproblem (Line 5). Thus, Algorithm 4 is a modified version of `Accelerated Extragradient`. To be consistent with the notation of the original paper, let

$$q(x) = f_1(x) + g(x), \quad p(x) = (f - f_1)(x).$$

We have

$$\|\nabla^2 p(x) - \nabla^2 p(y)\| \le \delta_f, \quad \forall x, y \in \mathbb{R}^d.$$

Moreover, Assumption 3 guarantees the convexity of $q$. This allows us to apply Theorem 1 from (Kovalev et al., 2022) with $\theta = 1/\delta_f$ and obtain $\mathcal{O}\left(\sqrt{\delta_f/\mu} \log 1/\varepsilon\right)$ communication rounds over only $M_f$ to achieve an arbitrary $\varepsilon$-solution. To guarantee the convergence of Algorithm 4, it is required to solve the subproblem in Line 5 with a certain accuracy:

$$\left\| A_\theta^k(\overline{x}_{k+1}) \right\|^2 \le \frac{\delta_f^2}{3} \left\| \underline{x}_k - \arg\min_{x \in \mathbb{R}^d} A_\theta^k(x) \right\|^2. \tag{6}$$

Unlike the original paper, computing $A_\theta^k(x)$ requires communication. This necessitates finding an efficient method to solve the subproblem 6. We can rewrite it as

$$A_\theta^k(x) = q_g(x) + p_g(x),$$

where

$$q_g(x) = \langle \nabla(f - f_1)(\underline{x}_k), x \rangle + \frac{1}{2\theta_f} \|x - \underline{x}_k\|^2 + f_1(x) + g_1(x), \quad p_g(x) = (g - g_1)(x).$$

Working with $q_g$ does not require communication. This pertains to the gradient sliding technique and suggests that $A_\theta^k$ can be minimized by using `Accelerated Extragradient` once more. We slightly modify the original proof and obtain linear convergence of Algorithm 5 by the norm of the gradient. This is important since equation 6 requires exactly this criterion. We now combine the obtained results. We formulate this as a corollary.

**Theorem 4.** *Consider Algorithm 4 for the problem 2 and Algorithm 5 for its subproblem 6. Then the complexities in terms of communication rounds are*

$$\mathcal{O}\left(\sqrt{\frac{\delta_f}{\mu}} \log \frac{1}{\varepsilon}\right), \quad \tilde{\mathcal{O}}\left(\sqrt{\frac{\delta_g}{\mu}} \log \frac{1}{\varepsilon}\right)$$

*for the nodes from $M_f$, $M_g$, respectively.*

---

**Algorithm 5** `AccExtragradient` for $A_\theta^k$

---

1: **Input:** $x_0 = \overline{x}_0 \in \mathbb{R}^d$
2: **Parameters:** $\tau_g \in (0,1)$, $\eta_g, \theta_g, \alpha_g, K > 0$
3: **for** $t = 0, 1, \ldots, T - 1$ **do**
4:      $\underline{x}_t = \tau_g x_t + (1 - \tau_g)\overline{x}_t$
5:      $\overline{x}_{t+1} \approx \arg\min_{x \in \mathbb{R}^d} \left[ B_{\theta_g}^t(x) \right]$, where

$$B_{\theta_g}^t(x) = \langle \nabla(g - g_1)(\underline{x}_t), x \rangle + \frac{1}{2\theta_g}\|x - \underline{x}_t\|^2$$

$$+ q_g(x)$$

6:      $x_{t+1} = x_t + \eta_g \alpha_g(\overline{x}_{t+1} - x_t) - \eta_g \nabla A_\theta^k(\overline{x}_{t+1})$
7: **end for**
8: **Output:** $\overline{x}_T$

---

See the proof in Appendix F.

## 7.1 DISCUSSION

Assumption on $g$ convexity allows us to construct an approach that achieves suboptimal complexity over $M_f$ and $M_g$ simultaneously. As mentioned earlier, without considering heterogeneity within the data distribution, the optimal method is `Accelerated Extragradient`. Applied to our setting, it yields $\tilde{\mathcal{O}}\left(\sqrt{(\delta_f + \delta_g)/\mu}\right)$ rounds over both $M_f$ and $M_g$. By complicating the structure of the problem and relying on real-world scenarios, we can break through this bound.

## 8 NUMERICAL EXPERIMENTS

Our theoretical insights are confirmed numerically on different classification tasks. We consider the distributed minimization of the negative cross-entropy:

$$h(x) = -\frac{1}{M} \sum_{m=1}^{M} \frac{1}{n_m} \sum_{j=1}^{n_m} \sum_{c \in C} y_{j,c}^m \log \hat{y}_{j,c}^m(a_j^m, x), \tag{7}$$

where $C$ is the set of classes, $y_{j,c}^m$ and $\hat{y}_{j,c}^m(a_j^m, x)$ are the $c$-th components of one-hot encoded and predicted label for the sample $a_j^m$, respectively. Motivated by the opportunity to introduce heterogeneity in the distribution of modes, we choose two sets of classes $(C_f, C_g)$ and create an imbalance between them in such a way that the server has more objects from $C_f$ than from $C_g$. Moreover, we divide the nodes (excepting the server) into two groups: $M_f$ and $M_g$, containing only $C_f$ and $C_g$, respectively. Thus, we aim to use $\delta_f < \delta_g$ to communicate with the fraction of the devices less frequently. In accordance with equation 2, the objective takes the form:

$$h(x) = f(x) + g(x) = -\frac{1}{|M_f|} \sum_{m \in M_f} \frac{1}{n_m} \sum_{j=1}^{n_m} \sum_{c \in C_f} y_{j,c}^m \log \hat{y}_{j,c}^m(a_j^m, x)$$

$$-\frac{1}{|M_g|} \sum_{m \in M_g} \frac{1}{n_m} \sum_{j=1}^{n_m} \sum_{c \in C_g} y_{j,c}^m \log \hat{y}_{j,c}^m(a_j^m, x). \tag{8}$$

In order to construct setups with different $\delta_g/\delta_f$ ratios, we introduce a disparity index $\kappa$, defined as the proportion of objects from $C_f$ among all available data on the server. Thus, $\kappa = 1$ means that it contains only $C_f$, and $\kappa = 1/2$ corresponds to a completely homogeneous scenario (equal $\delta_f$ and $\delta_g$). Since it is impossible to estimate $\delta_f$, $\delta_g$ analytically, we tune the parameters of each algorithm to the fastest convergence.

In this work, we provide a comparison of our approaches with distributed learning methods, such as `ProxyProx` (Woodworth et al., 2023), `Accelerated Extragradient` (AEG) (Kovalev et al., 2022).

## 8.1 MULTILAYER PERCEPTRON

Firstly, we use *MLP* to solve the *MNIST* (Deng, 2012) classification problem with $C_f = \{0, \ldots, 3\}$, $C_g = \{4, \ldots, 9\}$, $|M_f| = |M_g|$. To keep the task from being too simple, we consider the three-layer network (784, 64, 10 parameters).

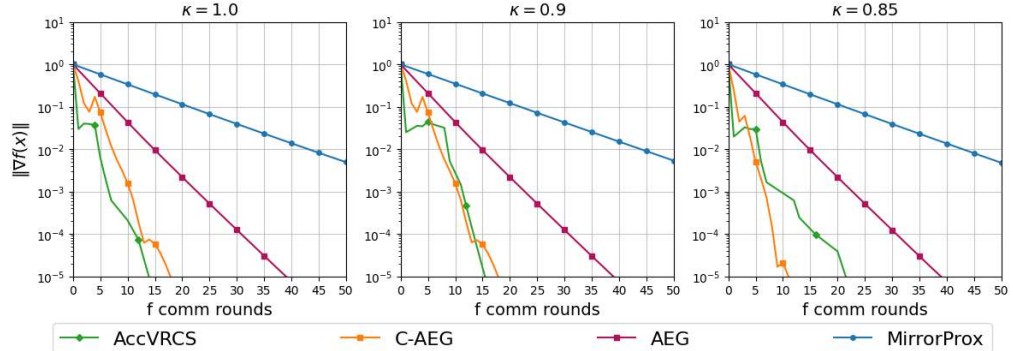

Figure 1: Comparison of state-of-the-art distributed methods on equation 8 with $|M_f| = |M_g| = 32$ and *MNIST* dataset. The criterion is the number of communication rounds over $M_f$. To show robustness, we vary the disparity parameter $\kappa$.

Figure 1 demonstrates clear superiority of the proposed approach in terms of communication with $M_f$. This effect is achieved through the dissimilar use of well- and poorly-conditioned clients. This experiment demonstrates the potential of complexity separation techniques in processing real-world federated learning scenarios, where the server represents different parts of the sample unevenly.

## 8.2 RESNET-18

In the second part of the experimental section, we consider *CIFAR-10* (Krizhevsky et al., 2009) with $C_f = \{4, \ldots, 9\}$, $C_g = \{0, \ldots, 3\}$, $|M_f| = |M_g|$. Since variance reduction in deep learning is associated with various challenges (Defazio and Bottou, 2019), we focus on comparing the two approaches: `SC-Extragradient` (Algorithm 1) and `Accelerated Extragradient` (Kovalev et al., 2022). To minimize equation 8, we implement two heads in *ResNet-18* (He et al., 2016), each corresponding to its respective set of classes. The weighted average classification accuracy for objects from $C_f$ and $C_g$ is used as a metric. The curves for the examined strategies are presented in Figure 2.

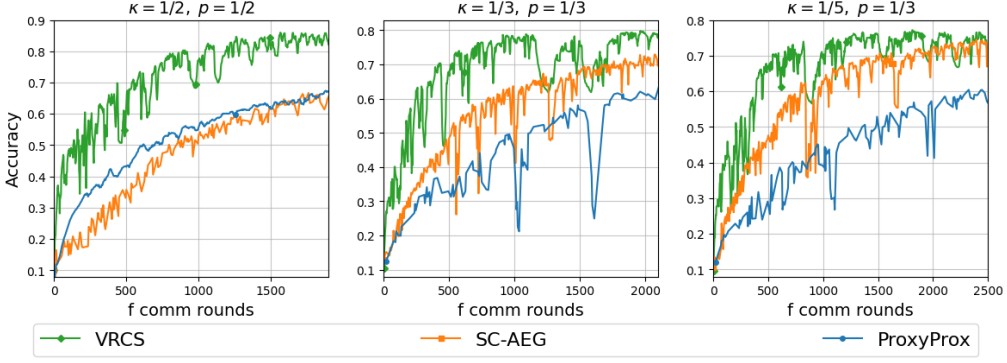

Figure 2: Comparison of `Accelerated Extragradient` and `SC-AccExtragradient` on equation 8 with $|M_f| = |M_g| = 5$ and *CIFAR-10* dataset. The criterion is the number of communication rounds over $M_f$. To show robustness, we vary the disparity parameter $\kappa$.

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

APPENDIX

CONTENTS

## A   AUXILLARY LEMMAS

**Lemma 1.** *(**Three-point equality**), (Acharyya et al., 2013) .   Given a differentiable function* $h\colon \mathbb{R}^d \to \mathbb{R}$. *We have*

$$\langle x - y, \nabla h(y) - \nabla h(z) \rangle = D_h(x, z) - D_h(x, y) - D_h(y, z).$$

**Lemma 2.** *(Allen-Zhu, 2018) Given a sequence* $D_0, D_1, ..., D_N \in \mathbb{R}$, *where* $N \in Geom(p)$. *Then*

$$\mathbb{E}_N[D_{N-1}] = pD_0 + (1 - p)\mathbb{E}_N[D_N].$$

**Lemma 3.** *(Allen-Zhu, 2018) If* $g$ *is proper* $\sigma$-*strongly convex and* $z_{k+1} = \arg\min_{z \in \mathbb{R}} \left[ \frac{1}{2\alpha}\|z - z_k\|^2 + \langle G_{k+1}, z \rangle + g(z) \right]$, *then for every* $x \in \mathbb{R}^d$ *we have*

$$\langle G_{k+1}, z_k - x \rangle + g(z_{k+1}) - g(x) \leq \frac{\alpha}{2}\|G_{k+1}\|^2 + \frac{\|z_k - x\|^2}{2\alpha} - \frac{(1 + \sigma\alpha)}{2\alpha}\|z_{k+1} - x\|^2.$$

## B   PROOF OF THEOREM 1

**Theorem 5.** *(Theorem 1) Consider Algorithm 1 for the problem 2 under Assumptions 1-4, , with the following tuning:*

$$\theta \leq \frac{1}{3(\delta_f + \delta_g)}, \quad \tau = \sqrt{\mu\theta}, \quad \eta = \min\left\{ \frac{1}{2\mu}, \frac{1}{2}\sqrt{\frac{\theta}{\mu}} \right\}, \quad \alpha = \mu. \tag{9}$$

*Let* $\overline{x}_{k+1}$ *satisfy:*

$$\mathbb{E}\left[ \|\nabla A_\theta^k(\overline{x}_{k+1})\|^2 \right] \leq \mathbb{E}\left[ \frac{\theta^2}{11}\|\underline{x}_k - \arg\min_{x \in \mathbb{R}^d} A_\theta^k(x)\|^2 \right].$$

*Then the complexities in terms of communication rounds are*

$$\mathcal{O}\left( \sqrt{\frac{\delta_f}{\mu}}\log\frac{1}{\varepsilon} + \frac{\sigma^2}{\mu\varepsilon} \right) \text{ for the nodes from } M_f,$$

*and*

$$\mathcal{O}\left( \sqrt{\frac{\delta_g}{\mu}}\log\frac{1}{\varepsilon} + \frac{\sigma^2}{\mu\varepsilon} \right) \text{ for the nodes from } M_g.$$

*Proof.* We begin with writing the norm of the argument in the standard way, as is usually done in convergence proofs:

$$\frac{1}{\eta}\|x_{k+1} - x_*\|^2 = \frac{1}{\eta}\|x_k - x_*\|^2 + \frac{2}{\eta}\langle x_{k+1} - x_k, x_k - x_* \rangle + \frac{1}{\eta}\|x_{k+1} - x_k\|^2.$$

Next, we expand the scalar product using Line 8 and obtain

$$\frac{1}{\eta}\|x_{k+1} - x_*\|^2 = \frac{1}{\eta}\|x_k - x_*\|^2 + 2\alpha\langle \overline{x}_{k+1} - x_k, x_k - x_* \rangle - 2\langle \zeta_k, x_k - x_* \rangle + \frac{1}{\eta}\|x_{k+1} - x_k\|^2$$

$$= \frac{1}{\eta}\|x_k - x_*\|^2 + 2\alpha\langle \overline{x}_{k+1} - x_k, x_k - x_* \rangle - 2\langle \zeta_k, x_k - x_* \rangle$$

$$+ 2\eta\alpha^2\|\overline{x}_{k+1} - x_k\|^2 + 2\eta\|\zeta_k\|^2.$$

On the right hand, we have only two terms depending on $\zeta_k$. The expectation over $\zeta_K$ of the scalar product is easy to take, since $x_k$, $x_*$ are independent of this random variable, and $\zeta_k$ itself gives an unbiased estimate of $\nabla h(\overline{x}_{k+1})$. We get

$$\mathbb{E}_{\zeta_k}\left[ \frac{1}{\eta}\|x_{k+1} - x_*\|^2 \right] = \frac{1}{\eta}\|x_k - x_*\|^2 + 2\alpha\langle \overline{x}_{k+1} - x_k, x_k - x_* \rangle - 2\langle \nabla h(\overline{x}_{k+1}), x_k - x_* \rangle$$

$$+ 2\eta\alpha^2\|\overline{x}_{k+1} - x_k\|^2 + 2\eta\mathbb{E}_{\zeta_k}\left[ \|\zeta_k\|^2 \right]. \tag{10}$$

To deal with $\mathbb{E}\left[\|\zeta_k\|^2\right]$, we use the smart zero technique, adding and subtracting $\nabla h(\overline{x}_{k+1})$. We have

$$
\begin{aligned}
\mathbb{E}_{\zeta_k}\left[\|\zeta_k\|^2\right] &= \mathbb{E}_{\zeta_k}\left[\|(\zeta_k - \nabla h(\overline{x}_{k+1})) + \nabla h(\overline{x}_{k+1})\|^2\right] \\
&= \mathbb{E}_{\zeta_k}\left[\|\zeta_k - \nabla h(\overline{x}_{k+1})\|^2\right] + \|\nabla h(\overline{x}_{k+1})\|^2 + 2\mathbb{E}_{\zeta_k}\left[\langle \zeta_k - \nabla h(\overline{x}_{k+1}), \nabla h(\overline{x}_{k+1})\rangle\right] \\
&= \mathbb{E}_{\zeta_k}\left[\|\zeta_k - \nabla h(\overline{x}_{k+1})\|^2\right] + \|\nabla h(\overline{x}_{k+1})\|^2.
\end{aligned}
$$

Here, the scalar product is zeroed because $\mathbb{E}_{\zeta_k}\left[\zeta_k\right] = \nabla h(\overline{x}_{k+1})$, and $\nabla h(\overline{x}_{k+1})$ is independent of $\zeta_k$. Now we are ready to use Assumption 4 and obtain

$$
\mathbb{E}_{\zeta_k}\left[\|\zeta_k\|^2\right] \leq \mathbb{E}\left[\|\nabla h(\overline{x}_{k+1})\|^2\right] + \sigma^2.
$$

Substitute this into equation 10 and get

$$
\begin{aligned}
\mathbb{E}_{\zeta_k}\left[\frac{1}{\eta}\|x_{k+1} - x_*\|^2\right] \leq &\frac{1}{\eta}\|x_k - x_*\|^2 + 2\alpha\langle \overline{x}_{k+1} - x_k, x_k - x_*\rangle - 2\langle \nabla h(\overline{x}_{k+1}), x_k - x_*\rangle \\
&+ 2\eta\alpha^2\|\overline{x}_{k+1} - x_k\|^2 + 2\eta\|\nabla h(\overline{x}_{k+1})\|^2 + 2\eta\sigma^2.
\end{aligned}
$$

Let us apply the formula for the square of the difference to $2\alpha\langle \overline{x}_{k+1} - x_k, x_k - x_*\rangle$. We obtain

$$
\begin{aligned}
\mathbb{E}_{\zeta_k}\left[\frac{1}{\eta}\|x_{k+1} - x_*\|^2\right] \leq &\frac{1 - \eta\alpha}{\eta}\|x_k - x_*\|^2 - \alpha\|\overline{x}_{k+1} - x_k\|^2 + \alpha\|\overline{x}_{k+1} - x_*\|^2 \\
&- 2\langle \nabla h(\overline{x}_{k+1}), x_k - x_*\rangle + 2\eta\alpha^2\|\overline{x}_{k+1} - x_k\|^2 + 2\eta\|\nabla h(\overline{x}_{k+1})\|^2 \\
&+ 2\eta\sigma^2.
\end{aligned}
$$

Using Line 4, we rewrite the last remaining scalar product and get

$$
\begin{aligned}
\mathbb{E}_{\zeta_k}\left[\frac{1}{\eta}\|x_{k+1} - x_*\|^2\right] \leq &\frac{1 - \eta\alpha}{\eta}\|x_k - x_*\|^2 - \alpha\|\overline{x}_{k+1} - x_k\|^2 + \alpha\|\overline{x}_{k+1} - x_*\|^2 \\
&+ 2\langle \nabla h(\overline{x}_{k+1}), x_* - \underline{x}_k\rangle + \frac{2(1-\tau)}{\tau}\langle \nabla h(\overline{x}_{k+1}), \overline{x}^k - \underline{x}_k\rangle \\
&+ 2\eta\alpha^2\|\overline{x}_{k+1} - x_k\|^2 + 2\eta\|\nabla h(\overline{x}_{k+1})\|^2 + 2\eta\sigma^2.
\end{aligned} \tag{11}
$$

To move on, we have to figure out what to do with the scalar product. Let us start with

$$
2\langle \nabla h(\overline{x}_{k+1}), x - \underline{x}_k\rangle = 2\langle \nabla h(\overline{x}_{k+1}), x - \overline{x}_{k+1}\rangle + 2\langle \nabla h(\overline{x}_{k+1}), \overline{x}_{k+1} - \underline{x}_k\rangle.
$$

In the first of the scalar products, we use strong convexity due to Assumption 1. We get

$$
2\langle \nabla h(\overline{x}_{k+1}), x - \underline{x}_k\rangle \leq [h(x) - h(\overline{x}_{k+1})] - \mu\|\overline{x}_{k+1} - x\|^2 + 2\theta\left\langle \nabla h(\overline{x}_{k+1}), \frac{\overline{x}_{k+1} - \underline{x}_k}{\theta}\right\rangle.
$$

Then, using the square of the difference once again, we obtain

$$
\begin{aligned}
2\langle \nabla h(\overline{x}_{k+1}), x - \underline{x}_k\rangle \leq &[h(x) - h(\overline{x}_{k+1})] - \mu\|\overline{x}_{k+1} - x\|^2 + 2\langle \nabla h(\overline{x}_{k+1}), \overline{x}_{k+1} - \underline{x}_k\rangle \\
=& [h(x) - h(\overline{x}_{k+1})] - \mu\|\overline{x}_{k+1} - x\|^2 - \frac{1}{\theta}\|\overline{x}_{k+1} - \underline{x}_k\|^2 - \theta\|\nabla h(\overline{x}_{k+1})\|^2 \\
&+ \theta\left\|\frac{\overline{x}_{k+1} - \underline{x}_k}{\theta} + \nabla h(\overline{x}_{k+1})\right\|^2.
\end{aligned}
$$

$$\tag{12}$$

The last expression on the right hand is almost $A_\theta^k(\overline{x}_{k+1})$ from Line 6. Let us take a closer look on it:

$$
\begin{aligned}
\left\|\frac{\overline{x}_{k+1} - \underline{x}_k}{\theta} + \nabla h(\overline{x}_{k+1})\right\|^2 &= \left\|\frac{\overline{x}_{k+1} - \underline{x}_k}{\theta} + \nabla(h - h_1)(\overline{x}_{k+1}) + \nabla h_1(\overline{x}_{k+1})\right\|^2 \\
&= \left\|\nabla A_\theta^k(\overline{x}_{k+1}) - \xi_k + \nabla(h - h_1)(\overline{x}_{k+1})\right\|^2 \\
&= \left\|\nabla A_\theta^k(\overline{x}_{k+1}) - \xi_k + \nabla(h - h_1)(\overline{x}_{k+1})\right\|^2 \\
&\leq 3\|\nabla A_\theta^k(\overline{x}_{k+1})\|^2 + 3\|\xi_k - \nabla(h - h_1)(\underline{x}_k)\|^2 \\
&\quad + 3\|\nabla(h - h_1)(\underline{x}_k) - \nabla(h - h_1)(\overline{x}_{k+1})\|^2.
\end{aligned}
$$

Using Assumption 2, we obtain

$$\left\|\frac{\overline{x}_{k+1} - \underline{x}_k}{\theta} + \nabla h(\overline{x}_{k+1})\right\|^2 \leq 3\|\nabla A_\theta^k(\overline{x}_{k+1})\|^2 + 3\|\xi_k - \nabla(h - h_1)(\underline{x}_k)\|^2$$
$$+ 3(\delta_f + \delta_g)^2 \|\overline{x}_{k+1} - \underline{x}_k\|^2.$$

Note that now the right-hand side depends on the random variable $\xi_k$. Using Assumption 4, we write the estimate for the mathematical expectation of the expression:

$$\mathbb{E}_{\xi_k}\left[\left\|\frac{\overline{x}_{k+1} - \underline{x}_k}{\theta} + \nabla h(\overline{x}_{k+1})\right\|^2\right] \leq \mathbb{E}_{\xi_k}\left[3\|\nabla A_\theta^k(\overline{x}_{k+1})\|^2 + 3(\delta_f + \delta_g)^2 \|\overline{x}_{k+1} - \underline{x}_k\|^2\right] + 3\sigma^2.$$
(13)

Substituting equation 13 into equation 12, we obtain

$$\mathbb{E}_{\xi_k}\left[2\langle\nabla h(\overline{x}_{k+1}), x - \underline{x}_k\rangle\right] \leq \mathbb{E}_{\xi_k}\Big[[h(x) - h(\overline{x}_{k+1})] - \mu\|\overline{x}_{k+1} - x\|^2$$
$$- \frac{1}{\theta}\left(1 - 3(\delta_f + \delta_g)^2\theta^2\right)\|\overline{x}_{k+1} - \underline{x}_k\|^2 - \theta\|\nabla h(\overline{x}_{k+1})\|^2$$
$$+ 3\theta\|A_\theta^k(\overline{x}_{k+1})\|^2\Big] + 3\theta\sigma^2.$$

Choosing $\theta \leq 1/3(\delta_f + \delta_g)$, we get

$$\mathbb{E}_{\xi_k}\left[2\langle\nabla h(\overline{x}_{k+1}), x - \underline{x}_k\rangle\right] \leq \mathbb{E}_{\xi_k}\Big[[h(x) - h(\overline{x}_{k+1})] - \mu\|\overline{x}_{k+1} - x\|^2 - \frac{2}{3\theta}\|\overline{x}_{k+1} - \underline{x}_k\|^2$$
$$- \theta\|\nabla h(\overline{x}_{k+1})\|^2 + 3\theta\|A_\theta^k(\overline{x}_{k+1})\|^2\Big] + 3\theta\sigma^2.$$

Note that

$$-\|a - b\|^2 \leq -\frac{1}{2}\|a - c\|^2 + \|b - c\|^2.$$

Thus, we have

$$\mathbb{E}_{\xi_k}\left[2\langle\nabla h(\overline{x}_{k+1}), x - \underline{x}_k\rangle\right] \leq \mathbb{E}_{\xi_k}\Big[[h(x) - h(\overline{x}_{k+1})] - \mu\|\overline{x}_{k+1} - x\|^2$$
$$- \frac{1}{3\theta}\|\underline{x}_k - \arg\min_{x\in\mathbb{R}^d} A_\theta^k(x)\|^2 + \frac{2}{3\theta}\|\overline{x}_{k+1} - \arg\min_{x\in\mathbb{R}^d} A_\theta^k(x)\|^2$$
$$- \theta\|\nabla h(\overline{x}_{k+1})\|^2 + 3\theta\|A_\theta^k(\overline{x}_{k+1})\|^2\Big]$$
$$+ 3\theta\sigma^2.$$

$A_\theta^k$ is $1/\theta$-strongly convex. This implies

$$\frac{2}{3\theta}\|\overline{x}_{k+1} - \arg\min_{x\in\mathbb{R}^d} A_\theta^k(x)\|^2 \leq \frac{2\theta}{3}\|\nabla A_\theta^k(\overline{x}_{k+1})\|^2$$

Hence, we can write

$$\mathbb{E}_{\xi_k}\left[2\langle\nabla h(\overline{x}_{k+1}), x - \underline{x}_k\rangle\right] \leq \mathbb{E}_{\xi_k}\Big[[h(x) - h(\overline{x}_{k+1})] - \mu\|\overline{x}_{k+1} - x\|^2$$
$$- \frac{1}{3\theta}\|\underline{x}_k - \arg\min_{x\in\mathbb{R}^d} A_\theta^k(x)\|^2 - \theta\|\nabla h(\overline{x}_{k+1})\|^2$$
$$+ \frac{11\theta}{3}\|A_\theta^k(\overline{x}_{k+1})\|^2\Big] + 3\theta\sigma^2.$$

Using equation 4, we conclude:

$$\mathbb{E}_{\xi_k}\left[2\langle\nabla h(\overline{x}_{k+1}), x - \underline{x}_k\rangle\right] \leq \mathbb{E}_{\xi_k}\Big[[h(x) - h(\overline{x}_{k+1})] - \mu\|\overline{x}_{k+1} - x\|^2 - \theta\|\nabla h(\overline{x}_{k+1})\|^2\Big]$$
$$+ 3\theta\sigma^2.$$
(14)

We take the expectation of equation 11 over $\xi_k$ and substitute equation 14. Taking $\alpha = \mu$ into account, we obtain

$$\mathbb{E}_{\zeta_k,\xi_k}\left[\frac{1}{\eta}\|x_k - x_*\|^2\right] \leq \mathbb{E}_{\zeta_k,\xi_k}\Big[\frac{1 - \eta\alpha}{\eta}\|x_k - x_*\|^2 - \alpha(1 - 2\eta\alpha)\|\overline{x}_{k+1} - x_k\|^2$$
$$+ 2\eta\|\nabla h(\overline{x}_{k+1})\|^2 + 2\eta\sigma^2 + [h(x_*) - h(\overline{x}_{k+1})]$$
$$+ \frac{1 - \tau}{\tau}[h(\overline{x}_k) - h(\overline{x}_{k+1})] - \frac{\theta}{\tau}\|\nabla h(\overline{x}_{k+1})\|^2 + \frac{3\theta}{\tau}\sigma^2\Big].$$

With our choice of parameters (see equation 9), we have

$$\mathbb{E}_{\zeta_k,\xi_k}\left[\frac{1}{\eta}\|x_k - x_*\|^2\right] \leq \mathbb{E}_{\zeta_k,\xi_k}\Big[\frac{1 - \eta\alpha}{\eta}\|x_k - x_*\|^2 + \frac{1}{\tau}[h(x_*) - h(\overline{x}_{k+1})]$$
$$+ \frac{1 - \tau}{\tau}[h(\overline{x}_k) - h(x_*)] + \frac{4\theta}{\tau}\sigma^2\Big].$$

Multiplying this expression by $\tau$, we obtain

$$\mathbb{E}_{\zeta_k,\xi_k}\left[\frac{\tau}{\eta}\|x_k - x_*\|^2 + [h(\overline{x}_{k+1}) - h(x_*)]\right] \leq \mathbb{E}_{\zeta_k,\xi_k}\Big[\frac{\tau}{\eta}(1 - \eta\alpha)\|x_k - x_*\|^2$$
$$+ (1 - \tau)[h(\overline{x}_k) - h(x_*)]\Big] + \frac{4\theta}{\tau}\sigma^2.$$

Denote

$$\Phi_k = \frac{\tau}{\eta}\|x_k - x_*\|^2 + [h(\overline{x}_k) - h(x_*)].$$

Using the choice of parameters as in equation 9, write down the result:

$$\mathbb{E}_{\zeta_k,\xi_k}[\Phi_{k+1}] \leq \left(1 - \frac{1}{2}\sqrt{\mu\theta}\right)\Phi_k + 4\theta\sigma^2.$$

Thus, we have convergence to some neighborhood of the solution. To achieve the "true" convergence, we have to make a finer tuning of $\theta$. Stich (2019) analyzed the recurrence sequence

$$0 \leq (1 - a\gamma)r_k - r_{k+1} + c\gamma^2, \quad \gamma \leq \frac{1}{d}$$

and obtained (see Lemma 2 in (Stich, 2019))

$$ar_{K+1} \leq \tilde{\mathcal{O}}\left(dr_0\exp\left\{-\frac{aK}{d}\right\} + \frac{c}{aK}\right).$$

In our analysis, we have

$$\gamma = \sqrt{\theta}, \quad d = \frac{1}{\sqrt{3(\delta_f + \delta_g)}}, \quad a = \frac{\sqrt{\mu}}{2}, \quad c = 4\sigma^2.$$

Thus, Algorithm 1 requires

$$\mathcal{O}\left(\sqrt{\frac{\delta_f + \delta_g}{\mu}}\log\frac{1}{\varepsilon} + \frac{\sigma^2}{\mu\varepsilon}\right) \text{ epochs}$$

to converge to an arbitrary $\varepsilon$-solution. Of these, the $p$ fraction engages only $M_f$ and the $1 - p$ uses only $M_g$. Choosing $p = \delta_f/(\delta_f + \delta_g)$ and using $\delta_f < \delta_g$, we obtain

$$\mathcal{O}\left(\sqrt{\frac{\delta_f}{\mu}}\log\frac{1}{\varepsilon} + \frac{\sigma^2}{\mu\varepsilon}\right) \text{ communication rounds for } M_f,$$

and

$$\mathcal{O}\left(\sqrt{\frac{\delta_g}{\mu}}\log\frac{1}{\varepsilon} + \frac{\sigma^2}{\mu\varepsilon}\right) \text{ communication rounds for } M_g.$$

$\square$

## C    DESCENT LEMMA FOR VARIANCE REDUCTION

**Lemma 4.** *Consider an epoch of Algorithm 2. Consider* $\psi(x) = h^1(x) - h(x) + 1/2\theta\|x\|^2$, *where* $\theta \leq 1/2(\delta_f + \delta_g)$. *Let* $x_{t+1}$ *satisfy*

$$\mathbb{E}\|\nabla A_\theta^t(x_{t+1})\|^2 \leq \frac{\mu}{17\theta} \left\| \arg\min_{x \in \mathbb{R}^d} A_\theta^t(x) - x_t \right\|^2.$$

*Then the following inequality holds for every* $x \in \mathbb{R}^d$:

$$\mathbb{E}\left[h(x_T) - h(x)\right] \leq \mathbb{E}\left[ qD_\psi(x, x_0) - qD_\psi(x, x_T) + 8\theta^2 \left( \frac{\delta_f^2}{p} + \frac{\delta_g^2}{1-p} \right) D_\psi(x_0, x_T) \right.$$
$$\left. - \frac{\mu\theta}{3} D_\psi(x, x_T) \right].$$

*Proof.* Let us differentiate the subproblem (Line 9):

$$\nabla A_\theta^t(x) = e_t + \frac{x - x_0}{\theta} + \nabla h_1(x).$$

After substituting $e_t$, we have

$$\nabla A_\theta^t(x) = \xi_t - \zeta_t + \frac{x - x_0}{\theta} + \nabla h_1(x).$$

Next, we add and subtract the expressions: $\nabla h(x), \nabla h(x_t), \nabla h_1(x_t)$. After grouping the terms, we get

$$\nabla A_\theta^t(x) = \{[\xi_t - \zeta_t] - [\nabla(h - h_1)(x_t) - \nabla(h - h_1)(x_0)]\}$$
$$+ \left\{\nabla(h_1 - h)(x) + \frac{x}{\theta} - \nabla(h_1 - h)(x_t) - \frac{x_t}{\theta}\right\} \tag{15}$$
$$+ \nabla h(x).$$

In the conditions of Lemma 4, we defined distance generating function as

$$\psi(x) = h_1(x) - h(x) + \frac{1}{2\theta}\|x\|^2.$$

It is not difficult to notice the presence of its gradient in equation 15. Thus, we have

$$\nabla A_\theta^t(x) = \{[\xi_t - \zeta_t] - [\nabla(h - h_1)(x_t) - \nabla(h - h_1)(x_0)]\}$$
$$+ \{\nabla\psi(x) - \nabla\psi(x_t)\} \tag{16}$$
$$+ \nabla h(x).$$

Now we can express $\nabla h(x)$. Using definition of strong convexity (Definition 3), we write

$$h(x_{t+1}) - h(x) \leq \langle x - x_{t+1}, -\nabla h(x_{t+1}) \rangle - \frac{\mu}{2}\|x_{t+1} - x\|^2.$$

Substituting equation 16, we obtain

$$h(x_{t+1}) - h(x) \leq \langle x - x_{t+1}, [\xi_t - \zeta_t] - [\nabla(h - h_1)(x_t) - \nabla(h - h_1)(x_0)] \rangle$$
$$+ \langle x - x_{t+1}, \nabla\psi(x_{t+1}) - \nabla\psi(x) \rangle - \langle x - x_{t+1}, \nabla A_\theta^t(x_{t+1}) \rangle$$
$$- \frac{\mu}{2}\|x_{t+1} - x\|^2.$$

Rewriting the first scalar product using smart zero $x_k$, we obtain

$$h(x_{t+1}) - h(x) \leq \langle x - x_t, [\xi_t - \zeta_t] - [\nabla(h - h_1)(x_t) - \nabla(h - h_1)(x_0)] \rangle$$
$$+ \langle x_t - x_{t+1}, [\xi_t - \zeta_t] - [\nabla(h - h_1)(x_t) - \nabla(h - h_1)(x_0)] \rangle$$
$$+ \langle x - x_{t+1}, \nabla\psi(x_{t+1}) - \nabla\psi(x) \rangle - \langle x - x_{t+1}, \nabla A_\theta^t(x_{t+1}) \rangle$$
$$- \frac{\mu}{2}\|x_{t+1} - x\|^2.$$

Let us apply Young's inequality to the second scalar product. We get

$$
\begin{aligned}
h(x_{t+1}) - h(x) \leq & \langle x - x_t, [\xi_t - \zeta_t] - [\nabla(h - h_1)(x_t) - \nabla(h - h_1)(x_0)]\rangle \\
& + \frac{1}{2\alpha}\|x_{t+1} - x_t\|^2 + \frac{\alpha}{2}\|[\xi_t - \zeta_t] - [\nabla(h - h_1)(x_t) - \nabla(h - h_1)(x_0)]\|^2 \\
& + \langle x - x_{t+1}, \nabla\psi(x_{t+1}) - \nabla\psi(x)\rangle - \langle x - x_{t+1}, \nabla A_\theta^t(x_{t+1})\rangle \\
& - \frac{\mu}{2}\|x_{t+1} - x\|^2.
\end{aligned}
$$

After that, we apply Young's inequality again, now to $\langle x - x_{t+1}, \nabla A_\theta^t(x_{t+1})\rangle$. This allows us to write

$$
\begin{aligned}
h(x_{t+1}) - h(x) \leq & \langle x - x_t, [\xi_t - \zeta_t] - [\nabla(h - h_1)(x_t) - \nabla(h - h_1)(x_0)]\rangle \\
& + \frac{1}{2\alpha}\|x_{t+1} - x_t\|^2 + \frac{\alpha}{2}\|[\xi_t - \zeta_t] - [\nabla(h - h_1)(x_t) - \nabla(h - h_1)(x_0)]\|^2 \\
& + \langle x - x_{t+1}, \nabla\psi(x_{t+1}) - \nabla\psi(x)\rangle + \frac{1}{\mu}\|\nabla A_\theta^t(x_{t+1})\|^2 - \frac{\mu}{4}\|x_{t+1} - x\|^2.
\end{aligned}
$$

Next, we use the three-point equality (Lemma 1) and obtain

$$
\begin{aligned}
h(x_{t+1}) - h(x) \leq & \langle x - x_t, [\xi_t - \zeta_t] - [\nabla(h - h_1)(x_t) - \nabla(h - h_1)(x_0)]\rangle \\
& + \frac{1}{2\alpha}\|x_{t+1} - x_t\|^2 + \frac{\alpha}{2}\|[\xi_t - \zeta_t] - [\nabla(h - h_1)(x_t) - \nabla(h - h_1)(x_0)]\|^2 \\
& + D_\psi(x, x_t) - D_\psi(x, x_{t+1}) - D_\psi(x_{t+1}, x_t) + \frac{1}{\mu}\|\nabla A_\theta^t(x_{t+1})\|^2 \\
& - \frac{\mu}{4}\|x_{t+1} - x\|^2.
\end{aligned}
\tag{17}
$$

Note that equation 17 contains expressions that depend on the choice between $f - f_1$ and $g - g_1$ ($i_k$). We get rid of it by passing to the mathematical expectation. Let us consider some terms of equation 17 separately. We note that

$$
\mathbb{E}_{i_t}\left[\langle x - x_t, [\xi_t - \zeta_t] - [\nabla(h - h_1)(x_t) - \nabla(h - h_1)(x_0)]\rangle\right] = 0, \tag{18}
$$

since $x$, $x_t$ are do not depend on $i_t$ and $\xi_t - \zeta_t$ is unbiased estimator of $\nabla(h - h_1)(x_t) - \nabla(h - h_1)(x_0)$ (see our explanations in the main text). Moreover, carefully looking at $\frac{\alpha}{2}\|[\xi_t - \zeta_t] - [\nabla(h - h_1)(x_t) - \nabla(h - h_1)(x_0)]\|^2$, we notice

$$
\begin{aligned}
& \mathbb{E}_{i_t}\left[\frac{\alpha}{2}\|[\xi_t - \zeta_t] - [\nabla(h - h_1)(x_t) - \nabla(h - h_1)(x_0)]\|^2\right] \\
\leq & \frac{\alpha}{2}\mathbb{E}_{i_t}\left[\|\xi_t - \zeta_t\|^2\right] \\
\leq & \frac{\alpha}{2}\frac{1}{p}\|\nabla(f - f_1)(x_t) - \nabla(f - f_1)(x_0)\|^2 \\
& + \frac{\alpha}{2}\frac{1}{1 - p}\|\nabla(g - g_1)(x_t) - \nabla(g - g_1(x_0)\|^2.
\end{aligned}
$$

Here, the first transition takes advantage of the fact that $\xi_t - \zeta_t$ estimates $\nabla(h - h_1)(x_t) - \nabla(h - h_1)(x_0)$ in the unbiased way. Given Hessian similarity (Assumption 2), this implies

$$
\mathbb{E}_{i_t}\left[\frac{\alpha}{2}\|[\xi_t - \zeta_t] - [\nabla(h - h_1)(x_t) - \nabla(h - h_1)(x_0)]\|^2\right] \leq \frac{\alpha}{2}\left(\frac{\delta_f^2}{p} + \frac{\delta_g^2}{1 - p}\right)\|x_t - x_0\|^2. \tag{19}
$$

Substituting equation 18 and equation 19 into equation 17, we obtain

$$
\begin{aligned}
\mathbb{E}_{i_t}[h(x_{t+1}) - h(x)] \leq & \mathbb{E}_{i_t}\Bigg[D_\psi(x, x_t) - D_\psi(x, x_{t+1}) - D_\psi(x_{t+1}, x_t) + \frac{1}{2\alpha}\|x_{t+1} - x_t\|^2 \\
& + \frac{\alpha}{2}\left(\frac{\delta_f^2}{p} + \frac{\delta_g^2}{1 - p}\right)\|x_t - x_0\|^2 + \frac{1}{\mu}\|\nabla A_\theta^t(x_{t+1})\|^2 - \frac{\mu}{4}\|x_{t+1} - x\|^2\Bigg].
\end{aligned}
\tag{20}
$$

Since $\theta \leq 1/2(\delta_f + \delta_g)$, it holds that

$$0 \leq \frac{1 - \theta/(\delta_f + \delta_g)}{2\theta} \|x - y\|^2 \leq D_\psi(x, y) \leq \frac{1 + \theta/(\delta_f + \delta_g)}{2\theta} \|x - y\|^2, \quad \forall x, y \in \mathbb{R}^d. \qquad (21)$$

Thus, we can estimate

$$-D_\psi(x_{t+1}, x_t) \leq -\frac{1 - \theta/(\delta_f + \delta_g)}{2\theta} \|x_{t+1} - x_t\|^2.$$

Substituting it into equation 20 and taking $\alpha = \frac{2\theta}{1 - \theta/(\delta_f + \delta_g)}$, we get

$$\mathbb{E}_{i_t} \left[ h(x_{t+1}) - h(x) \right] \leq \mathbb{E}_{i_t} \left[ D_\psi(x, x_t) - D_\psi(x, x_{t+1}) - \frac{1 - \theta/(\delta_f + \delta_g)}{4\theta} \|x_{t+1} - x_t\|^2 \right.$$

$$+ \frac{1}{\mu} \|\nabla A_\theta^t(x_{t+1})\|^2 + \frac{\theta}{1 - \theta/(\delta_f + \delta_g)} \left( \frac{\delta_f^2}{p} + \frac{\delta_g^2}{1 - p} \right) \|x_t - x_0\|^2$$

$$\left. - \frac{\mu}{4} \|x_{t+1} - x\|^2 \right].$$

Since $\theta \leq 1/2(\delta_f + \delta_g)$, we have

$$-\frac{1 - \theta/\delta_f + \delta_g}{4\theta} \|x_{t+1} - x_t\|^2 \leq -\frac{1}{8\theta} \|x_{t+1} - x_t\|^2.$$

Further, we note that

$$-\|a - b\|^2 \leq -\frac{1}{2} \|a - c\|^2 + \|b - c\|^2.$$

Combining all the remarks, we obtain

$$\mathbb{E}_{i_t} \left[ h(x_{t+1}) - h(x) \right] \leq \mathbb{E}_{i_t} \left[ D_\psi(x, x_t) - D_\psi(x, x_{t+1}) + \frac{\theta}{1 - \theta/(\delta_f + \delta_g)} \left( \frac{\delta_f^2}{p} + \frac{\delta_g^2}{1 - p} \right) \|x_t - x_0\|^2 \right.$$

$$+ \frac{1}{\mu} \|\nabla A_\theta^t(x_{t+1})\|^2 - \frac{1}{16\theta} \|x_t - \arg\min_{x \in \mathbb{R}^d} A_\theta^t(x)\|^2$$

$$\left. + \frac{1}{8\theta} \|x_{t+1} - \arg\min_{x \in \mathbb{R}^d} A_\theta^t(x)\|^2 - \frac{\mu}{4} \|x_{t+1} - x\|^2 \right].$$

$$(22)$$

Let us look carefully at the second row of the expression. Since $A_\theta^t$ is $1/\theta$-strongly convex, it holds that

$$\frac{1}{8\theta} \|x_{t+1} - \arg\min_{x \in \mathbb{R}^d} A_\theta^t(x)\|^2 \leq \frac{\theta}{8} \|\nabla A_\theta^t(x_{t+1})\|^2.$$

Thus,

$$\frac{1}{\mu} \|\nabla A_\theta^t(x_{t+1})\|^2 - \frac{1}{16\theta} \|x_t - \arg\min_{x \in \mathbb{R}^d} A_\theta^t(x)\|^2 + \frac{1}{8\theta} \|x_{t+1} - \arg\min_{x \in \mathbb{R}^d} A_\theta^t(x)\|^2$$

$$\leq \frac{8 + \theta\mu}{8\mu} \left[ \|\nabla A_\theta^t(x_{t+1})\|^2 - \frac{\mu}{2\theta(\mu\theta + 8)} \|x_t - \arg\min_{x \in \mathbb{R}^d} A_\theta^t(x)\|^2 \right]$$

$$\leq \frac{8 + \theta\mu}{8\mu} \left[ \|\nabla A_\theta^t(x_{t+1})\|^2 - \frac{\mu}{17\theta} \|x_t - \arg\min_{x \in \mathbb{R}^d} A_\theta^t(x)\|^2 \right].$$

Taking equation 5 into account, we get rid of this term in the obtained estimate. We rewrite equation 22 as

$$\mathbb{E}_{i_t}\left[h(x_{t+1}) - h(x)\right] \leq \mathbb{E}_{i_t}\left[ D_\psi(x, x_t) - D_\psi(x, x_{t+1}) \right.$$

$$+ \frac{\theta}{1 - \theta/(\delta_f + \delta_g)}\left(\frac{\delta_f^2}{p} + \frac{\delta_g^2}{1-p}\right)\|x_t - x_0\|^2$$

$$\left. - \frac{\mu}{4}\|x_{t+1} - x\|^2\right].$$

For $(T-1)$-th iteration we have

$$\mathbb{E}_{i_{T-1}}\left[h(x_T) - h(x)\right] \leq \mathbb{E}_{i_{T-1}}\left[ D_\psi(x, x_{T-1}) - D_\psi(x, x_T) \right.$$

$$+ \frac{\theta}{1 - \theta/(\delta_f + \delta_g)}\left(\frac{\delta_f^2}{p} + \frac{\delta_g^2}{1-p}\right)\|x_{T-1} - x_0\|^2$$

$$\left. - \frac{\mu}{4}\|x_T - x\|^2\right].$$

As discussed above, $T-1$ is the geometrically distributed random variable. Thus, we can write the mathematical expectation by this quantity as well and use the tower-property. We have

$$\mathbb{E}\left[h(x_T) - h(x)\right] \leq \mathbb{E}\left[ D_\psi(x, x_{T-1}) - D_\psi(x, x_T) \right.$$

$$+ \frac{\theta}{1 - \theta/(\delta_f + \delta_g)}\left(\frac{\delta_f^2}{p} + \frac{\delta_g^2}{1-p}\right)\|x_{T-1} - x_0\|^2$$

$$\left. - \frac{\mu}{4}\|x_T - x\|^2\right].$$

Using Lemma 2, we obtain

$$\mathbb{E}\left[h(x_T) - h(x)\right] \leq \mathbb{E}\left[ D_\psi(x, x_0) - D_\psi(x, x_T) + \frac{\theta}{1 - \theta/(\delta_f + \delta_g)}\left(\frac{\delta_f^2}{p} + \frac{\delta_g^2}{1-p}\right)\|x_T - x_0\|^2 \right.$$

$$\left. - \frac{\mu}{4}\|x_T - x\|^2\right].$$

Taking $\theta \leq 1/2(\delta_f + \delta_g)$ and equation 21 into account, we write

$$\mathbb{E}\left[h(x_T) - h(x)\right] \leq \mathbb{E}\left[ qD_\psi(x, x_0) - qD_\psi(x, x_T) + 8\theta^2\left(\frac{\delta_f^2}{p} + \frac{\delta_g^2}{1-p}\right)D_\psi(x_0, x_T) \right.$$

$$\left. - \frac{\mu\theta}{3}D_\psi(x, x_T)\right].$$

This is the required. $\qquad\square$

## D  PROOF OF THEOREM 2

Now we are ready to prove the convergence of VRCS. Let us repeat the statement.

**Theorem 6.** *(**Theorem 2**) Consider Algorithm 6 for the problem 2 under Assumptions 1-2 and the conditions of Lemma 4, with the following tuning:*

$$\theta = \frac{1}{4}\sqrt{\frac{p(1-p)q}{p\delta_g^2 + (1-p)\delta_f^2}}, \quad p = q = \frac{\delta_f^2}{\delta_f^2 + \delta_g^2}. \tag{23}$$

---

**Algorithm 6** VRCS

---

1: **Input:** $x_0 \in \mathbb{R}^d$
2: **Parameters:** $p, q \in (0, 1)$, $\theta > 0$
3: **for** $k = 0, \ldots, K - 1$ **do**
4: $\quad x_{k+1} = \text{VRCS}^{\text{1ep}}(p, q, \theta, x_k)$
5: **end for**
6: **Output:** $x_K$

---

*Then the complexities in terms of communication rounds are*

$$\mathcal{O}\left(\frac{\delta_f}{\mu} \log \frac{1}{\varepsilon}\right) \text{ for the nodes from } M_f,$$

*and*

$$\mathcal{O}\left(\left(\frac{\delta_g}{\delta_f}\right) \frac{\delta_g}{\mu} \log \frac{1}{\varepsilon}\right) \text{ for the nodes from } M_g.$$

*Proof.* Let us apply Lemma 4 twice (Note that $D_\psi(x_k, x_k) = 0$):

$$\mathbb{E}\left[h(x_{k+1}) - h(x_*)\right] \leq \mathbb{E}\left[qD_\psi(x_*, x_k) - qD_\psi(x_*, x_{k+1}) + 8\theta^2\left(\frac{\delta_f^2}{p} + \frac{\delta_g^2}{1-p}\right)D_\psi(x_k, x_{k+1})\right.$$

$$\left. - \frac{\mu\theta}{3}D_\psi(x_*, x_{k+1})\right],$$

$$\mathbb{E}\left[h(x_{k+1}) - h(x_k)\right] \leq \mathbb{E}\left[-qD_\psi(x_k, x_{k+1}) + 8\theta^2\left(\frac{\delta_f^2}{p} + \frac{\delta_g^2}{1-p}\right)D_\psi(x_k, x_{k+1})\right.$$

$$\left. - \frac{\mu\theta}{3}D_\psi(x_k, x_{k+1})\right],$$

We note that $-\frac{\mu\theta}{3}D_\psi(x_k, x_{k+1}) \leq 0$ due to the strong convexity of $\psi$ (see equation 21). Summing up the above inequalities, we obtain

$$\mathbb{E}\left[2h(x_{k+1}) - h(x_k) - h(x_*)\right] \leq \mathbb{E}\left[qD_\psi(x_*, x_k) - \left(q + \frac{\mu\theta}{3}\right)D_\psi(x_*, x_{k+1})\right.$$

$$\left. + \left(16\theta^2\left(\frac{\delta_f^2}{p} + \frac{\delta_g^2}{1-p}\right) - q\right)D_\psi(x_k, x_{k+1})\right].$$

We have to get rid of $D_\xi(x_k, x_{k+1})$. Thus, we have to fine-tune $\theta$ as

$$\theta \leq \frac{\sqrt{p(1-p)q}}{4\sqrt{p\delta_g^2 + (1-p)\delta_f^2}}.$$

Thus, we have

$$\theta = \min\left\{\frac{1}{2(\delta_f + \delta_g)}, \frac{\sqrt{p(1-p)q}}{4\sqrt{p\delta_g^2 + (1-p)\delta_f^2}}\right\}.$$

With choive of parameters given in equation 23, we have

$$\frac{\sqrt{p(1-p)q}}{4\sqrt{p\delta_g^2 + (1-p)\delta_f^2}} \leq \frac{1}{2(\delta_f + \delta_g)},$$

which indeed allows us to consider

$$\theta = \frac{\sqrt{p(1-p)q}}{4\sqrt{p\delta_g^2 + (1-p)\delta_f^2}}.$$

Thus, we have

$$\mathbb{E}\left[[h(x_{k+1}) - h(x_*)] + q\left(1 + \frac{\mu\theta}{3q}\right)D_\psi(x_*, x_{k+1})\right] \leq \mathbb{E}\left[qD_\psi(x_*, x_k) + \frac{1}{2}[h(x_k) - h(x_*)]\right].$$

With our choice of parameters (see equation 23), we can note

$$\left(1 + \frac{\mu\theta}{3q}\right)^{-1} \leq 1 - \frac{\mu\theta}{6q}$$

and conclude that Algorithm 6 requires

$$\tilde{\mathcal{O}}\left(\frac{q}{\mu\theta}\right) \text{ iterations}$$

to achieve an arbitrary $\varepsilon$-solution. Iteration of VRCS consists of the communication across all devices and then the epoch, at each iteration of which only $M_f$ or $M_g$ is engaged. The round length is on average $1/q$. Thus, VRCS requires

$$\tilde{\mathcal{O}}\left(\frac{q}{\mu\theta}\left(1 + \frac{p}{q}\right)\right) \text{ rounds for } M_f,$$

and

$$\tilde{\mathcal{O}}\left(\frac{q}{\mu\theta}\left(1 + \frac{1-p}{q}\right)\right) \text{ rounds for } M_g.$$

With our choice of parameters (see 23) we have

$$\tilde{\mathcal{O}}\left(\frac{\delta_f}{\mu}\right) \text{ rounds for } M_f,$$

and

$$\tilde{\mathcal{O}}\left(\left(\frac{\delta_g}{\delta_f}\right)\frac{\delta_g}{\mu}\right) \text{ rounds for } M_g.$$

$\square$

**Remark 1.** *The analysis of Algorithm 6 allows different complexities to be obtained, thus allowing adaptation to the parameters of a particular problem. For example, by varying $p$ and $q$, one can get $\tilde{\mathcal{O}}\left(\frac{\delta_g}{\mu}\right)$, $\tilde{\mathcal{O}}\left(\frac{\delta_g}{\mu}\right)$ or $\tilde{\mathcal{O}}\left(\frac{\sqrt{\delta_f\delta_g}}{\mu}\right)$, $\tilde{\mathcal{O}}\left(\frac{\sqrt{\delta_f\delta_g}}{\mu}\right)$. Unfortunately, it is not possible to obtain $\tilde{\mathcal{O}}\left(\frac{\delta_f}{\mu}\right)$ over $M_f$ and $\tilde{\mathcal{O}}\left(\frac{\delta_g}{\mu}\right)$ over $M_g$ simultaneously.*

# E    PROOF OF THEOREM 3

**Theorem 7.** *(**Theorem 3**) Consider Algorithm 3 for the problem 2 under Assumptions 1-2 and the conditions of Lemma 4, with the following tuning:*

$$\theta = \frac{1}{4}\sqrt{\frac{p(1-p)q}{p\delta_g^2 + (1-p)\delta_f^2}}, \quad \tau = \sqrt{\frac{\theta\mu}{3q}}, \quad \alpha = \sqrt{\frac{\theta}{3\mu q}}, \quad p = q = \frac{\delta_f^2}{\delta_f^2 + \delta_g^2}. \tag{24}$$

*Then the complexities in terms of communication rounds are*

$$\mathcal{O}\left(\sqrt{\frac{\delta_f}{\mu}}\log\frac{1}{\varepsilon}\right) \text{ for the nodes from } M_f,$$

*and*

$$\mathcal{O}\left(\left(\frac{\delta_g}{\delta_f}\right)^{3/2}\sqrt{\frac{\delta_g}{\mu}}\log\frac{1}{\varepsilon}\right) \text{ for the nodes from } M_g.$$

*Proof.* We start with Lemma 4, given that $y_{k+1} = \texttt{VRCS}^{\texttt{1ep}}(p, q, \theta, x_{k+1})$. Let us write

$$\mathbb{E}\left[h(y_{k+1}) - h(x)\right] \leq \mathbb{E}\left[qD_\psi(x, x_{k+1}) - qD_\psi(x, y_{k+1}) + 8\theta^2\left(\frac{\delta_f^2}{p} + \frac{\delta_g^2}{1-p}\right)D_\psi(x_{k+1}, y_{k+1})\right.$$

$$\left. - \frac{\mu}{4}\|y_{k+1} - x\|^2\right].$$

(25)

Using three-point equality (see Lemma 1), we note

$$qD_\psi(x, x_{k+1}) - qD_\psi(x, y_{k+1}) = q\langle x - x_{k+1}, \nabla\psi(y_{k+1}) - \nabla\psi(x_{k+1})\rangle - qD_\psi(x_{k+1}, y_{k+1}).$$

Substituting it into equation 25, we obtain

$$\mathbb{E}\left[h(y_{k+1}) - h(x)\right] \leq \mathbb{E}\left[q\langle x - x_{k+1}, \nabla\psi(y_{k+1}) - \nabla\psi(x_{k+1})\rangle - \frac{\mu}{4}\|y_{k+1} - x\|^2\right.$$

$$\left. + \left\{8\theta^2\left(\frac{\delta_f^2}{p} + \frac{\delta_g^2}{1-p}\right) - q\right\}D_\psi(x_{k+1}, y_{k+1})\right].$$

With our choice of $\theta$ (see equation 24), we have

$$\left\{8\theta^2\left(\frac{\delta_f^2}{p} + \frac{\delta_g^2}{1-p}\right) - q\right\}D_\psi(x_{k+1}, y_{k+1}) \leq -\frac{q}{2}D_\psi(x_{k+1}, y_{k+1}).$$

Thus, we can write

$$\mathbb{E}\left[h(y_{k+1}) - h(x)\right] \leq \mathbb{E}\left[q\langle x - x_{k+1}, \nabla\psi(y_{k+1}) - \nabla\psi(x_{k+1})\rangle - \frac{\mu}{4}\|y_{k+1} - x\|^2\right.$$

$$\left. - \frac{q}{2}D_\psi(x_{k+1}, y_{k+1})\right].$$

We suggest to add and subtract $z_k$ in the scalar product to get

$$\mathbb{E}\left[h(y_{k+1}) - h(x)\right] \leq \mathbb{E}\left[q\langle x - z_k, \nabla\psi(y_{k+1}) - \nabla\psi(x_{k+1})\rangle\right.$$

$$+ q\langle z_k - x_{k+1}, \nabla\psi(y_{k+1}) - \nabla\psi(x_{k+1})\rangle$$

(26)

$$\left. - \frac{\mu}{4}\|y_{k+1} - x\|^2 - \frac{q}{2}D_\psi(x_{k+1}, y_{k+1})\right].$$

Looking carefully at Line 4, we note that

$$z_k - x_{k+1} = \frac{1-\tau}{\tau}(x_{k+1} - y_k).$$

Substituting it into equation 26, we get

$$\mathbb{E}\left[h(y_{k+1}) - h(x)\right] \leq \mathbb{E}\left[q\langle x - z_k, \nabla\psi(y_{k+1}) - \nabla\psi(x_{k+1})\rangle\right.$$

$$+ \frac{1-\tau}{\tau}q\langle x_{k+1} - y_k, \nabla\psi(y_{k+1}) - \nabla\psi(x_{k+1})\rangle$$

(27)

$$\left. - \frac{\mu}{4}\|y_{k+1} - x\|^2 - \frac{q}{2}D_\psi(x_{k+1}, y_{k+1})\right].$$

Next, let us analyze $q\langle x_{k+1} - y_k, \nabla\psi(y_{k+1}) - \nabla\psi(x_{k+1})\rangle$. It is not difficult to see that this term appears in Lemma 4 if we substitute $x = y_k$. Writing it out, we obtain

$$\mathbb{E}\left[q\langle x_{k+1} - y_k, \nabla\psi(y_{k+1}) - \nabla\psi(x_{k+1})\rangle\right] \leq \mathbb{E}\left[[h(y_k) - h(y_{k+1})] - \frac{q}{2}D_\psi(x_{k+1}, y_{k+1})\right].$$

Plugging this into equation 27, we derive

$$\mathbb{E}\left[h(y_{k+1}) - h(x)\right] \leq \mathbb{E}\left[q\langle x - z_k, \nabla\psi(y_{k+1}) - \nabla\psi(x_{k+1})\rangle + \frac{1-\tau}{\tau}[h(y_k) - h(y_{k+1})]\right.$$

$$\left. - \frac{\mu}{4}\|y_{k+1} - x\|^2 - \frac{q}{2\tau}D_\psi(x_{k+1}, y_{k+1})\right].$$

Note that $G_{k+1} = q(\nabla\psi(x_{k+1}) - \nabla\psi(y_{k+1}))$. Thus, we have

$$\mathbb{E}\left[h(y_{k+1}) - h(x)\right] \leq \mathbb{E}\left[\langle z_k - x, G_{k+1}\rangle + \frac{1-\tau}{\tau}[h(y_k) - h(y_{k+1})] - \frac{\mu}{4}\|y_{k+1} - x\|^2\right.$$

$$\left. - \frac{q}{2\tau}D_\psi(x_{k+1}, y_{k+1})\right]. \tag{28}$$

Next, we apply Lemma 3 to Line 8 in order to evaluate

$$\langle z_k - x, G_{k+1}\rangle - \frac{\mu}{4}\|y_{k+1} - x\|^2 \leq \langle z_k - x, G_{k+1}\rangle - \frac{\mu}{4}\|y_{k+1} - x\|^2 + \frac{\mu}{4}\|y_{k+1} - z_{k+1}\|^2$$

$$\leq \frac{\alpha}{2}\|G_{k+1}\|^2 + \frac{1}{2\alpha}\|z_k - x\|^2 + \frac{1 + 0.5\alpha}{2\alpha}\|z_{k+1} - x\|^2. \tag{29}$$

We also have to estimate $\|G_{k+1}\|^2$.

$$\|G_{k+1}\|^2 \leq q^2\|\nabla\psi(x_{k+1}) - \nabla\psi(y_{k+1})\|^2 \leq q^2\frac{2(1 + \theta(\delta_f + \delta_g))}{\theta}D_\psi(x_{k+1}, y_{k+1})$$

$$\leq \frac{3q^2}{\theta}D_\psi(x_{k+1}, y_{k+1}). \tag{30}$$

Substituting equation 30 and equation 29 into equation 28, we conclude

$$\mathbb{E}\left[\frac{1}{\tau}[h(y_{k+1}) - h(x)] + \frac{1 + 0.5\mu\alpha}{2\alpha}\|z_{k+1} - x\|^2\right] \leq \mathbb{E}\left[\frac{1-\tau}{\tau}[h(y_k) - h(x)] + \frac{1}{2\alpha}\|z_k - x\|^2\right.$$

$$\left. - \frac{q}{2\tau}\left(1 - \frac{3\alpha q\tau}{\theta}\right)D_\psi(x_{k+1}, y_{k+1})\right].$$

With our choice of parameters (see equation 24), we have

$$1 - \frac{3\alpha q\tau}{\theta} = 0$$

Moreover, $\mu\alpha < 1$ (with our choice of $\alpha$) and therefore,

$$\left(1 + \frac{\mu\alpha}{2}\right)^{-1} \leq 1 - \frac{\mu\alpha}{4}.$$

Thus, we conclude that Algorithm 3 requires

$$\tilde{\mathcal{O}}\left(\sqrt{\frac{q}{\theta\mu}}\right) \text{ iterations}$$

to achieve an arbitrary $\varepsilon$-solution. The same as in Algorithm 2, iteration of `AccVRCS` consists of the communication across all devices and then the epoch with random choice of $M_f$ or $M_g$. Thus, `AccVRCS` requires

$$\tilde{\mathcal{O}}\left(\sqrt{\frac{q}{\mu\theta}}\left(1 + \frac{p}{q}\right)\right) \text{ rounds for } M_f,$$

and

$$\tilde{\mathcal{O}}\left(\sqrt{\frac{q}{\mu\theta}}\left(1 + \frac{1-p}{q}\right)\right) \text{ rounds for } M_g.$$

After substituting equation 24, this results in

$$\tilde{\mathcal{O}}\left(\sqrt{\frac{\delta_f}{\mu}}\right) \text{ rounds for } M_f,$$

and

$$\tilde{\mathcal{O}}\left(\left(\frac{\delta_g}{\delta_f}\right)^{3/2}\sqrt{\frac{\delta_g}{\mu}}\right) \text{ rounds for } M_g.$$

$\square$

## F    PROOF OF THEOREM 4

**Theorem 8.** *Consider Algorithm 5 for the problem 6 under Assumptions 2-3, with the following tuning:*

$$\theta_g \leq \frac{1}{2\delta_g}, \quad \tau_g = \frac{1}{2}\sqrt{\frac{\theta_g}{\theta}}, \quad \alpha_g = \frac{1}{\theta}, \quad \eta_g = \min\left\{\frac{\theta}{2}, \frac{1}{4}\frac{\theta_g}{\tau_g}\right\}; \tag{31}$$

*and let $\overline{x}_{t+1}$ satisfy:*

$$\left\|B^t_{\theta_g}(\overline{x}_{t+1})\right\|^2 \leq \frac{2}{10\theta_g^2}\left\|\underline{x}_t - \arg\min_{x\in\mathbb{R}^d}B^t_{\theta_g}(x)\right\|^2.$$

*Then it takes*

$$\mathcal{O}\left(\sqrt{\theta\delta_g}\log\frac{1}{\varepsilon}\right) \text{ communication rounds}$$

*over only $M_g$ to achieve $\|\nabla A^k_{\theta_f}(\overline{x}_{t+1})\|^2 \leq \varepsilon$.*

*Proof.* The proof is much the same as the proof of Theorem 1 (see Appendix B). Nevertheless, we give it in the full form. We start with

$$\frac{1}{\eta_g}\|x_{t+1} - x_*\|^2 = \frac{1}{\eta_g}\|x_t - x_*\|^2 + \frac{2}{\eta_g}\langle x_{t+1} - x_t, x_t - x_*\rangle + \frac{1}{\eta_g}\|x_{t+1} - x_t\|^2.$$

Next, we use Line 6 to obtain

$$\frac{1}{\eta_g}\|x_{t+1} - x_*\|^2 = \frac{1}{\eta_g}\|x_t - x_*\|^2 + 2\alpha_g\langle\overline{x}_{t+1} - x_t, x_t - x_*\rangle + 2\langle\nabla A^k_\theta(\overline{x}_{t+1}), x_t - x_*\rangle$$

$$+ \frac{1}{\eta_g}\|x_{t+1} - x_t\|^2.$$

After that, we apply the formula for square of difference to the first scalar product and get

$$\frac{1}{\eta_g}\|x_{t+1} - x_*\|^2 = \frac{1}{\eta_g}\|x_t - x_*\|^2 + \alpha_g\|\overline{x}_{t+1} - x_*\|^2 - \alpha_g\|\overline{x}_{t+1} - x_t\|^2$$

$$- \alpha_g\|x_t - x_*\|^2 + 2\langle\nabla A^k_\theta(\overline{x}_{t+1}), x_t - x_*\rangle + \frac{1}{\eta_g}\|x_{t+1} - x_t\|^2.$$

Let us take a closer look at the last norm. Using Line 6, we obtain

$$\frac{1}{\eta_g}\|x_{t+1} - x_t\|^2 \leq 2\eta_g\alpha_g^2\|\overline{x}_{t+1} - x_t\|^2 + 2\eta\|\nabla A^k_\theta(\overline{x}_{t+1})\|^2.$$

Taking the choice of parameters (see equation 31) into account, we can write

$$\frac{1}{\eta_g}\|x_{t+1} - x_*\|^2 \leq \frac{1 - \eta_g\alpha_g}{\eta_g}\|x_t - x_*\|^2 + \alpha_g\|\overline{x}_{t+1} - x_*\|^2 + 2\langle\nabla A^k_\theta(\overline{x}_{t+1}), x_t - x_*\rangle$$

$$+ 2\eta_g\|\nabla A^k_\theta(\overline{x}_{k+1})\|^2.$$

It remains to use Line 4 to write out the remaining scalar product. We have

$$\frac{1}{\eta_g}\|x_{t+1} - x_*\|^2 \leq \frac{1 - \eta_g\alpha_g}{\eta_g}\|x_t - x_*\|^2 + \alpha_g\|\overline{x}_{t+1} - x_*\|^2 + 2\langle\nabla A^k_\theta(\overline{x}_{t+1}), x_* - \underline{x}_t\rangle$$

$$+ \frac{2(1 - \tau_g)}{\tau_g}\langle\nabla A^k_\theta(\overline{x}_{t+1}), \overline{x}_t - \underline{x}_t\rangle + 2\eta_g\|\nabla A^k_\theta(\overline{x}_{k+1})\|^2. \tag{32}$$

As in the proof of Theorem 1, we have to deal with the scalar products. Let us write

$$\langle\nabla A^k_\theta(\overline{x}_{t+1}), x - \underline{x}_t\rangle = \langle\nabla A^k_\theta(\overline{x}_{t+1}), x - \overline{x}_{t+1}\rangle + \langle\nabla A^k_\theta(\overline{x}_{t+1}), \overline{x}_{t+1} - \underline{x}_t\rangle.$$

We have already mentioned in the main text that $A_\theta^k$ is $1/\theta$-strongly convex. Indeed, $\nabla^2 A_\theta^k(x) \succeq \frac{1}{\theta}$. Thus, we can apply the definition of strong convexity (see Assumption 1) to the first scalar product:

$$\langle \nabla A_\theta^k(\overline{x}_{t+1}), x - \underline{x}_t \rangle \leq [A_\theta^k(\overline{x}_{t+1}) - A_\theta^k(x)] - \frac{1}{\theta}\|\overline{x}_{t+1} - x\|^2 + \theta_g \left\langle \nabla A_\theta^k(\overline{x}_{t+1}), \frac{\overline{x}_{t+1} - \underline{x}_t}{\theta_g} \right\rangle.$$

Next, we write out the remaining scalar product, exploiting the square of the difference, and obtain

$$\langle \nabla A_\theta^k(\overline{x}_{t+1}), x - \underline{x}_t \rangle \leq [A_\theta^k(\overline{x}_{t+1}) - A_\theta^k(x)] - \frac{1}{\theta}\|\overline{x}_{t+1} - x\|^2 - \theta_g\|\nabla A_\theta^k(\overline{x}_{t+1})\|^2$$
$$- \frac{1}{\theta_g}\|\overline{x}_{t+1} - \underline{x}_t\|^2 + \theta_g \left\|\nabla A_\theta^k(\overline{x}_{t+1}) + \frac{\overline{x}_{t+1} - \underline{x}_t}{\theta_g}\right\|^2. \tag{33}$$

Let us look carefully at the last norm and notice

$$\left\|\nabla A_\theta^k(\overline{x}_{t+1}) + \frac{\overline{x}_{t+1} - \underline{x}_t}{\theta_g}\right\|^2 = \left\|\nabla q_g(\overline{x}_{t+1}) + \nabla(g - g_1)(\overline{x}_{t+1}) + \frac{\overline{x}_{t+1} - \underline{x}_t}{\theta_g}\right\|^2$$
$$= \left\|\nabla B_{\theta_g}^t(\overline{x}_{t+1}) + \nabla(g - g_1)(\overline{x}_{t+1}) - \nabla(g - g_1)(\underline{x}_t)\right\|^2$$
$$\leq 2\|\nabla B_{\theta_g}^t(\overline{x}_{t+1})\|^2 + 2\|\nabla(g - g_1)(\overline{x}_{t+1}) - \nabla(g - g_1)(\underline{x}_t)\|^2.$$

Next, we apply the Hessian similarity (see Definition 1) and obtain

$$\left\|\nabla A_\theta^k(\overline{x}_{t+1}) + \frac{\overline{x}_{t+1} - \underline{x}_t}{\theta_g}\right\|^2 \leq 2\|\nabla B_{\theta_g}^t(\overline{x}_{t+1})\|^2 + 2\delta_g^2\|\overline{x}_{t+1} - \underline{x}_t\|^2 \tag{34}$$

Substituting equation 34 into equation 33, we get

$$\langle \nabla A_\theta^k(\overline{x}_{t+1}), x - \underline{x}_t \rangle \leq [A_\theta^k(\overline{x}_{t+1}) - A_\theta^k(x)] - \frac{1}{\theta}\|\overline{x}_{t+1} - x\|^2 - \theta_g\|\nabla A_\theta^k(\overline{x}_{t+1})\|^2$$
$$+ 2\theta_g\|\nabla B_{\theta_g}^t(\overline{x}_{t+1})\|^2 - \frac{1}{\theta_g}\left(1 - 2\theta_g^2\delta_g^2\right)\|\overline{x}_{t+1} - \underline{x}_t\|^2.$$

With $\theta \leq 1/2\theta_g$ (see equation 31), we have

$$\langle \nabla A_\theta^k(\overline{x}_{t+1}), x - \underline{x}_t \rangle \leq [A_\theta^k(\overline{x}_{t+1}) - A_\theta^k(x)] - \frac{1}{\theta}\|\overline{x}_{t+1} - x\|^2 - \theta_g\|\nabla A_\theta^k(\overline{x}_{t+1})\|^2$$
$$+ 2\theta_g\|\nabla B_{\theta_g}^t(\overline{x}_{t+1})\|^2 - \frac{1}{2\theta_g}\|\overline{x}_{t+1} - \underline{x}_t\|^2.$$

Note that

$$-\|a - b\|^2 \leq -\frac{1}{2}\|a - c\|^2 + \|b - c\|^2.$$

Thus, we can write

$$\langle \nabla A_\theta^k(\overline{x}_{t+1}), x - \underline{x}_t \rangle \leq [A_\theta^k(\overline{x}_{t+1}) - A_\theta^k(x)] - \frac{1}{\theta}\|\overline{x}_{t+1} - x\|^2 - \theta_g\|\nabla A_\theta^k(\overline{x}_{t+1})\|^2$$
$$+ 2\theta_g\|\nabla B_{\theta_g}^t(\overline{x}_{t+1})\|^2 - \frac{1}{4\theta_g}\|\underline{x}_t - \arg\min_{x \in \mathbb{R}^d} B_{\theta_g}^t(x)\|^2$$
$$+ \frac{1}{2\theta_g}\|\overline{x}_{t+1} - \arg\min_{x \in \mathbb{R}^d} B_{\theta_g}^t(x)\|^2.$$

$B_{\theta_g}^t$ is $1/\theta_g$-strongly convex. This implies

$$\langle \nabla A_\theta^k(\overline{x}_{t+1}), x - \underline{x}_t \rangle \leq [A_\theta^k(\overline{x}_{t+1}) - A_\theta^k(x)] - \frac{1}{\theta}\|\overline{x}_{t+1} - x\|^2 - \theta_g\|\nabla A_\theta^k(\overline{x}_{t+1})\|^2$$
$$+ \frac{5\theta_g}{2}\|\nabla B_{\theta_g}^t(\overline{x}_{t+1})\|^2 - \frac{1}{4\theta_g}\|\underline{x}_t - \arg\min_{x \in \mathbb{R}^d} B_{\theta_g}^t(x)\|^2.$$

The criterion helps to eliminate the last two terms. We conclude

$$\langle \nabla A_\theta^k(\overline{x}_{t+1}), x - \underline{x}_t \rangle \leq [A_\theta^k(\overline{x}_{t+1}) - A_\theta^k(x)] - \frac{1}{\theta}\|\overline{x}_{t+1} - x\|^2 - \theta_g\|\nabla A_\theta^k(\overline{x}_{t+1})\|^2.$$

We are ready to estimate the scalar products in equation 32. Let us write

$$\frac{1}{\eta_g}\|x_{t+1} - x_*\|^2 \leq \frac{1 - \eta_g \alpha_g}{\eta_g}\|x_t - x_*\|^2 + \left(\alpha_g - \frac{1}{\theta}\right)\|\overline{x}_{t+1} - x_*\|^2$$

$$+ \left(2\eta_g - \frac{\theta_g}{\tau_g}\right)\|\nabla A^k_{\theta_g}(\overline{x}_{t+1})\|^2$$

$$+ [A^k_\theta(\overline{x}_{t+1}) - A^k_\theta(x_*)] + \frac{1 - \tau_g}{\tau_g}[A^k_\theta(\overline{x}_{t+1}) - A^k_\theta(\overline{x}_t)].$$

Denote $\Phi_k = \frac{1}{\eta_g}\|x_t - x_*\|^2 + \frac{1}{\tau_g}[A^k_\theta(\overline{x}_t) - A^k_\theta(x_*)]$ With the proposed choice of parameters (see equation 31), we have

$$\Phi_{k+1} + \frac{\theta_g}{2\tau_g}\|\nabla A^k_\theta(\overline{x}_{k+1})\|^2 \leq \left(1 - \frac{1}{2}\sqrt{\frac{\theta_g}{\theta}}\right)\Phi_k$$

$$\leq \left(1 - \frac{1}{2}\sqrt{\frac{\theta_g}{\theta}}\right)\left[\Phi_k + \frac{\theta_g}{2\tau_g}\|\nabla A^k_\theta(\overline{x}_t)\|^2\right].$$

Rolling out the recursion and noting that $\frac{\theta_g}{2\tau_g}\|\nabla A^k_\theta(\overline{x}_K)\|^2 \leq \Phi_K + \frac{\theta_g}{2\tau_g}\|\nabla A^k_\theta(\overline{x}_K)\|^2$, we obtain linear convergence of Algorithm 5 by the gradient norm. It requires

$$\mathcal{O}\left(\sqrt{\theta\delta_g}\log\frac{1}{\varepsilon}\right) \text{ rounds over only } M_g$$

to converge to an arbitrary $\varepsilon$-solution. $\qquad\square$

## THE USE OF LARGE LANGUAGE MODELS (LLMS)

Language models were used to improve text quality (mostly to correct grammatical errors). LLMs were not used to obtain theoretical results or write code.

