# OpenReview forum: "Accelerated Methods with Complexity Separation Under Data Similarity for Federated Learning Problems"
_ICLR.cc/2026/Conference — Submitted to ICLR 2026_

### Official Review · Reviewer_RPoo · 2025-10-24

**Soundness:** 3
**Presentation:** 3
**Contribution:** 2
**Rating:** 4
**Confidence:** 3

**Summary:**

This paper proposes a set of accelerated methods for federated learning to improve communication efficiency. The core approach is to exploit data similarity, as measured by the Hessian of the loss functions. The framework partitions clients into two distinct groups based on their data similarity to the server, allowing for different communication frequencies to be applied to each group, thereby reducing overall communication overhead. The authors provide theoretical analysis for their algorithms and validate them on standard image classification datasets.

**Strengths:**

Strengths:
1.The paper addresses the communication bottleneck in federated learning, which is a significant and highly relevant problem, particularly in the context of non-IID data.
2.The proposed methods are accompanied by a formal theoretical analysis of their convergence properties and communication complexity, providing a solid foundation for the algorithmic design.
3.The idea of separating clients based on data similarity to apply different communication strategies is intuitive and logically sound as a potential way to optimize resource usage.

**Weaknesses:**

Weaknesses:
1.Incremental Novelty and Lack of Direct Comparison: The idea of leveraging data similarity to improve communication efficiency in federated learning is not new, a point the authors acknowledge by citing a body of related work such as (Lin et al., 2024), (Jiang et al., 2024), (Shamir et al., 2014), (Hendrikx et al., 2020), and (Kovalev et al., 2022). The paper's distinction—relying only on server-side similarity (h₁) rather than all-client similarity (hₘ)—appears to be an incremental step. The paper lacks a direct or intuitive comparison to demonstrate why this specific relaxation of assumptions leads to a significant advantage over the works it cites.
2.Idealized Client Partitioning Model: The strict partitioning of clients into two disjoint sets, Mf and Mg, is a significant idealization. In real-world scenarios, a single client's dataset will likely contain a mix of common and rare data patterns. The paper does not address how this partitioning would be implemented in practice or how the model would perform in more realistic data distributions, questioning its practical applicability.
3.Potential Contradiction in Theory and Experiments: The theoretical analysis for the C-AEG algorithm requires the function g to be convex. However, this algorithm is experimentally validated on a non-convex MLP model. This discrepancy should be addressed to ensure logical and theoretical consistency.
4.Insufficient Experimental Validation: The experiments are conducted on relatively standard datasets (MNIST and CIFAR-10). To make a more compelling case, the proposed methods should be tested in larger-scale, more complex federated scenarios that are more representative of real-world challenges.
5.Omission of Key Baselines: While the paper includes comparisons with some relevant baselines like AEG, it omits other prominent methods for handling data heterogeneity, such as Scaffold and FedProx. Including these would provide a more comprehensive assessment of the proposed methods' advantages.

**Questions:**

1.Could you elaborate on the practical, demonstrable advantages of relaxing the similarity assumption to only involve the server (h₁) compared to the cited works that assume similarity for all clients (hₘ)?
2.How would the proposed client partitioning scheme be applied in practice where a client’s local data may contain a mix of common and rare patterns? How robust is the model to this more realistic scenario?
3.Could you clarify the apparent discrepancy between the theoretical requirement of convexity for function g in the C-AEG analysis and its application to a non-convex MLP in the experiments? What are the implications of this for the validity of the results?
4.Why were key baselines designed to handle data heterogeneity, such as Scaffold and FedProx, not included in the experimental comparison?

---

### Official Review · Reviewer_jQKA · 2025-10-29

**Soundness:** 4
**Presentation:** 3
**Contribution:** 4
**Rating:** 6
**Confidence:** 2

**Summary:**

The paper examines **communication-complexity separation** in federated learning  with a **composite objective** $h=f+g$ under Hessian similarity where $\delta_f < \delta_g$. It develops three directions: (1) a stochastic-approximation + extrapolation baseline (**SC-AccExtragradient**) that first separates the round complexities of $M_f$ and $M_g$;  (2) **VRCS** achieves $\tilde{\mathcal{O}}(\delta_f / \mu)$ rounds on $M_f$ and $\tilde{\mathcal{O}}((\delta_g/\delta_f) \cdot \delta_g /u)$ on $M_g$; its accelerated variant **AccVRCS** achieves $\mathcal{O}(\sqrt{\delta_f / \mu} \log(1/\epsilon))$ on $M_f$ and $\mathcal{O}((\delta_g / \delta_f)^{3/2}\sqrt{\delta_g / \mu} \log(1/\epsilon))$ on $M_g$; and (3) with $g$ convex, a two-level accelerated extragradient scheme yielding near-optimal separation $\tilde{\mathcal{O}}(\sqrt{\delta_f /\mu})$ and $\tilde{\mathcal{O}}(\sqrt{\delta_g /\mu})$. Experiments on **MNIST-MLP** and **CIFAR-10-ResNet18** illustrate advantages in $M_f$ communication rounds.

**Strengths:**

1. The paper brings together the **composite structure** $h=f+g$ and and **Hessian similarity**, explicitly distinguishing $\delta_f$ and $\delta_g$ and using this split to guide sampling (and scheduling). This constitutes a natural yet underexplored extension of the “similarity + acceleration/variance-reduction” literature.
2.  Starting from a strong-convexity baseline, it first provides an upper bound with separated complexities (Thm. 1), then applies variance-reduction ideas to tighten the rates (Thm. 2–3), and finally, reaches (near) optimal bounds on both sides when $g$ is convex (Thm. 4).

**Weaknesses:**

1. The experiments are limited to MNIST-MLP and CIFAR-10-ResNet18, and primarily report the number of communication rounds on the $M_f$ side. The paper **does not report** (i) metrics on the $M_g$ side and (ii)  total communication volume (e.g., counts of vector exchanges). It also lacks evaluations on more **realistic federated settings** (standard FL benchmarks). These omissions restrict the external validity and practical relevance of the claims.

2. The manuscript uses multiple, partially inconsistent names and symbols for (accelerated) extrapolation and variance-reduction methods, which increases reader burden. A unified notation table and consistent algorithm names across text, figures, and pseudocode are **currently missing**. For example, it refers to “SC-AccExtragradient (Algorithm 1)” and captions Fig. 2 as “SC-AccExtragradient,” yet the surrounding text contrasts it with “SC-Extragradient.” Likewise, in Algorithm 4 (**C-AccExtragradient**), the header declares generic parameters $\tau, \eta, \cdots$ (no subscripts), whereas the body immediately switches to $\tau_f, \eta_f, \cdots$.

**Questions:**

See above weakness.

---

### Official Review · Reviewer_4cxz · 2025-11-06

**Soundness:** 2
**Presentation:** 1
**Contribution:** 1
**Rating:** 0
**Confidence:** 4

**Summary:**

The paper carefully formulates federated learning optimization under data heterogeneity as a composite optimization problem with two components reflecting common and rare data modes. Under the Hessian similarity assumption, the authors develop and analyze several algorithms based on stochastic gradient descent, variance reduction, and extragradient methods that achieve separated communication complexity bounds for the two client groups.

**Strengths:**

The work extends prior literature on Hessian similarity and complexity separation to the setting where two separate data modes have different similarity conditions.

**Weaknesses:**

1. The core contribution appears somewhat unjustified and seems like a convenient simplification. The authors propose decomposing the client functions into two composite parts representing “common” and “rare” data modes distributed over different client sets. However, a natural question arises: why restrict the heterogeneity to only two partitions? Since client heterogeneity is often modeled by continuous distributions (e.g., Dirichlet), segmenting into just two groups feels like a simplistic shortcut aimed at applying existing composite optimization techniques rather than addressing realistic data heterogeneity.

2. The theory sections (Sections 5-7) lacks organization and feels quite fragmented. Instead of presenting a unified analytical framework, the paper introduces five different algorithms addressing SGD, VR, VR with acceleration, and extragradients, without fully synthesizing these results cohesively. Additionally, many terms (such as what is “good” in line 329) are left vague or undefined, which detracts from the theoretical rigor. Much of the analysis reads as a straightforward extension of existing results under the composite assumption (assumption 2) and lacks genuinely novel theoretical insights or algorithmic innovations. If the work aims to repackage known techniques under a novel assumption that itself is insufficiently argued, the paper's contribution is being overstated.

3. The manuscript gives the impression of a rushed submission with numerous writing quality issues scattered throughout Sections 5-8, encompassing both theory and experiments. There are multiple errors, unclear explanations, and poor presentation that significantly reduce readability and accessibility. This lack of polish suggests the work is not yet ready for publication and requires substantial revision.

**Questions:**

Apart from the weaknesses, I would like to point out some other issues of the paper:

1. It seems like algorithm 1 should be "SC-Extragradient" instead of "SC-AccExtragradient" since acceleration is proposed at algorithm 4. This error has been across the whole paper and causes huge confusion when trying to understand the specific algorithm that they are refering to.

2. I think the "CS" in the "VRCS" algorithm stands for "complexity seperation"? Then why is algorithm 1 named "SC-Extragradient" instead of  ""CS-Extragradient""?

3. All 5 algorithms have completely different naming strategies, some with abbreviations and some without, in the experiments the name SC-AEG also came out of nowhere, I'M think it should be algorithm 4 but there is no explanation on it.

4. Line 8 of Algorithm 2 is nothing like the equation written at the beginning of section 6, and the term $w_0$ is not defined properly.

5. Section 8.2 lacks any discussion regarding the experimental results.

6. The results in Section 8.2 exhibit inexplicable discrepancies: there is no comparison with the Accelerated Extragradient (AEG) algorithm, whereas "ProxyProx" appears on the plots without any mentioning in section 8.2. Moreover, the plots feature Algorithms 2 and 4, despite the text claiming comparisons involving Algorithm 1, which itself is absent from these figures.

**Details Of Ethics Concerns:**

No concerns.

---

### Official Review · Reviewer_WjqF · 2025-11-10

**Soundness:** 3
**Presentation:** 1
**Contribution:** 3
**Rating:** 4
**Confidence:** 2

**Summary:**

This paper examines the communication efficiency of federated learning systems under data similarity. The authors consider a setting in which client data is heterogeneous, thereby degrading the quality of existing methods that exploit data similarity. Grounded in the observation that clients can be considered as either containing frequent or rare modes, a composite objective is formulated. Under this lens, the authors derive several algorithms that, under varying assumptions, achieve improved communication complexities.

It is a very interesting approach, and the methodology is numerically verified on MNIST and CIFAR-10, where performance gains over existing methods across different data distribution scenarios are reported.

**Strengths:**

1. The paper solves a quite relevant and more realistic problem in federated learning setups when considering real data.

1. The paper is developed rigorously and is well-positioned in the existing state-of-the-art.

1. The paper splits up the optimization problem into two functions corresponding to frequent and rare data, and uses the Hessian similarity of the two functions to develop communication-efficient methods.

1. The paper also derives the complexities of the methods in terms of communication rounds.

**Weaknesses:**

1. The paper states (line 162) that the theory is validated through experiments on a "diverse set of tasks". This seems to be an overstatement, since the experiments are only conducted on two image classification tasks on MNIST and CIFAR-10.

1. The paper is quite challenging to read, with both prior and present results being described in the running text. I strongly encourage the authors to make the text more readable, as that will (probably) also increase the impact of their work. I find the ideas presented in the work very nice, but the presentation is lacking.

1. There is no conclusion in the paper. It ends with numerical experiments. This should be added.

**Questions:**

1. Line 96, why do you use $\nabla f$ for the gradient of f, but $g'$ for the gradient of $g$, assuming $\partial g$ means the gradient (or subgradient) of $g$?

1. Line 101: "VI"'s are not defined

1. Line 202: "normalize it by probability of choice." Could you elaborate on this?

1. Line 209, why are both expectations bounded by the same variance $\sigma_2$? Should these be two different numbers, or is it important that they are the same?

1. Line 224: Extra line should be removed

1. Figure 1: $\|  \nabla f(x)\|$ increases for some communication rounds for AccVRCS. Is this expected?

1. Are the results in Figures 1+2 based on one training pass, or are they averaged over multiple simulations? I expect this is only one run, due to the volatility of Figure 2.

---

### Meta-Review · Area_Chair_BftT · 2026-01-07

**Summary:**

The paper formulated federated learning optimization under the data heterogeneity as a composite optimization problem with two components reflecting common and rare data modes. Then it proposed a class of communication-efficient federated learning algorithms based on stochastic gradient descent, variance reduction, and extragradient methods under the Hessian similarity assumption. It provided  convergence analysis of the proposed algorithms. It also provided some simple experiments to demonstrate effectiveness of the proposed algorithms.

Although the authors provided a new view to handle data heterogeneity in federated learning, there exist many concerns from the reviewers. Since the authors do not give rebuttals to deal with any concerns from the reviewers, I suggest the authors follow the reviewers' suggestions to revise the paper to submit another conference or journal.

**Reviewer Concerns:**

The authors do not provide rebuttals to deal with any concerns from the reviewers.

**Reviewer Scores:**

Since the authors do not address any concerns of reviewers, clearly the reviewers could not change their scores.

---

### Decision · Program_Chairs · 2026-01-26

Reject